# The hypoxia-response pathway modulates RAS/MAPK–mediated cell fate decisions in *Caenorhabditis elegans*

Sabrina Maxeiner[1,2], Judith Grolleman[3], Tobias Schmid[1], Jan Kammenga[3], Alex Hajnal[1]

Animals need to adjust many cellular functions to oxygen availability to adapt to changing environmental conditions. We have used the nematode *Caenorhabditis elegans* as a model to investigate how variations in oxygen concentrations affect cell fate specification during development. Here, we show that several processes controlled by the conserved RTK/RAS/MAPK pathway are sensitive to changes in the atmospheric oxygen concentration. In the vulval precursor cells (VPCs), the hypoxia-inducible factor HIF-1 activates the expression of the nuclear hormone receptor NHR-57 to counteract RAS/MAPK–induced differentiation. Furthermore, cross-talk between the NOTCH and hypoxia-response pathways modulates the capability of the VPCs to respond to RAS/MAPK signaling. Lateral NOTCH signaling positively regulates the prolyl hydroxylase EGL-9, which promotes HIF-1 degradation in uncommitted VPCs and permits RAS/MAPK–induced differentiation. By inducing DELTA family NOTCH ligands, RAS/MAPK signaling creates a positive feedback loop that represses HIF-1 and NHR-57 expression in the proximal VPCs and keeps them capable of differentiating. This regulatory network formed by the NOTCH, hypoxia, and RAS/MAPK pathways may allow the animals to adapt developmental processes to variations in oxygen concentration.

## Introduction

The RAS/MAPK pathway regulates cell growth, differentiation, proliferation, apoptosis, and migration in all metazoans (Simanshu et al, 2017). Constitutively activating mutations in HRAS, NRAS, or KRAS are among the most frequent tumor-initiating mutations in human cancer. In *Caenorhabditis elegans*, RAS/MAPK signaling is involved in several developmental processes, such as the specification of the excretory duct cell precursor, the differentiation and maturation of meiotic germ cells or the development of the hermaphrodite vulva (Sternberg, 2005; Sundaram, 2006). During vulval development, an EGF-like ligand (LIN-3) secreted by the gonadal anchor cell (AC) activates EGFR/RAS/MAPK signaling in the adjacent vulval precursor cells (VPCs) through the EGF receptor (LET-23) pathway (Fig 1A). Strong activation of the RAS/MAPK signaling pathway in P6.p, the VPC located closest to the AC, induces the primary (1°) cell fate and up-regulates the expression of DELTA-like DSL ligands, which activate lateral NOTCH signaling in the adjacent VPCs (Chen & Greenwald, 2004). NOTCH performs two distinct functions during vulval induction (VI). First, before and after the first VPC division, NOTCH signaling maintains the VPCs competent, that is, capable of responding to the inductive AC signal (Wang & Sternberg, 1999). Second, strong NOTCH activation in P5.p and P7.p, which are the neighbors of P6.p, inhibits MAPK signaling and directly specifies the secondary (2°) fate (Berset et al, 2001; Yoo et al, 2004). The distal VPCs (P3-4.p and P8.p) that receive neither the EGFR/RAS/MAPK nor NOTCH signal adopt the tertiary (3°), uninduced fate. As a result, an invariant 3°-3°-2°-1°-2°-3° cell fate pattern is established. Activating (gain-of-function) mutations in the RAS/MAPK pathway lead to the ectopic induction of distal VPCs and a multivulva (Muv) phenotype, whereas loss-of-function mutations in RAS/MAPK pathway components cause a vulvaless (Vul) phenotype. Thus, the average number of induced VPCs per animal, termed the VI index, can be used to quantify RAS/MAPK signaling strength (e.g., VI = 3 in the wild-type, VI < 3 in Vul, and VI > 3 in Muv mutants) (Schmid et al, 2015).

*C. elegans* has evolved cellular and behavioral responses to adapt to variations in oxygen concentration (Semenza, 2001; Gray et al, 2004). At the cellular level, the hypoxia-response pathway mediates the adaptation to low oxygen conditions and a switch from aerobic to anaerobic metabolism. In ambient oxygen, the *C. elegans* hypoxia-inducible factor HIF-1 $\alpha$ is hydroxylated by the prolyl hydroxylase EGL-9 at a specific proline residue within the degradation domain (Fig 2A) (Epstein et al, 2001). Hydroxylated HIF-1 interacts with the von Hippel-Lindau E3 ubiquitin ligase VHL-1

[1]Institute of Molecular Life Sciences, University of Zurich, Zurich, Switzerland   [2]PhD Program in Molecular Life Sciences, University and ETH Zurich, Zurich, Switzerland   [3]Laboratory of Nematology, Wageningen University, Wageningen, The Netherlands

Correspondence: alex.hajnal@imls.uzh.ch
Judith Grolleman's present address is Radboud University Nijmegen Medical Centre, Department of Human Genetics, Nijmegen, The Netherlands

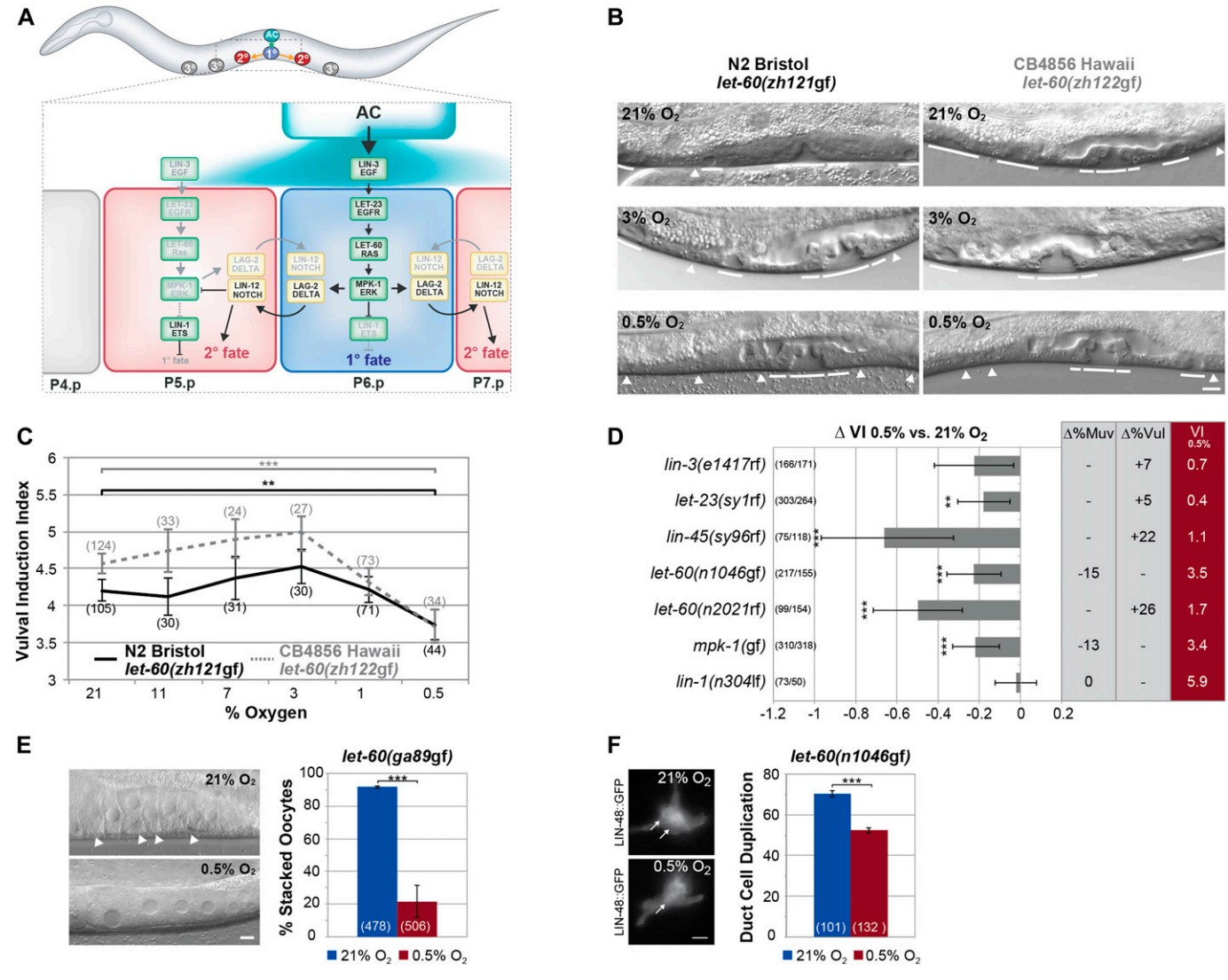

**Figure 1. Hypoxia represses RAS/MAPK–mediated differentiation in different tissues.**
**(A)** Overview of vulval development showing the known interactions between the RAS/MAPK and DELTA/NOTCH pathways. During the L2 stage, LIN-3 EGF activates the RAS/MAPK cascade in P5.p-P7.p. P6.p, in which RAS/MAPK activity is highest, adopts the 1° cell fate, and expresses DSL NOTCH ligands to activate LIN-12 NOTCH signaling in the adjacent Pn.p cells, which adopt the 2° fate. The uninduced P(3-4).p and P8.p cells adopt the 3° fate and fuse with the surrounding hypodermis. **(B)** Vulval phenotypes of the *let-60 ras* G13E mutation in the N2 Bristol (left) and CB4856 Hawaii (right) background with varying oxygen concentrations. Solid lines indicate induced 1° and 2° and arrowheads uninduced 3° VPCs in L4 larvae. **(C)** VI of N2 Bristol and CB4856 Hawaii *let-60 ras* G13E mutants raised in varying oxygen concentrations. **(D)** Effect of hypoxia on different RTK/RAS/MAPK pathway, *bar-1(ga80)*, *lin-12(n137)*, and *lin-12(n137n720)* mutants. ΔVI indicates the change in VI of animals raised in 0.5% compared with controls grown in 21% oxygen. Δ%Muv and Δ%Vul indicate the change in the percentage of animals with VI > 3 and VI < 3, respectively. The absolute VIs at 0.5% oxygen are shown in the rightmost red column. **(E)** Suppression of the stacked oocyte phenotype in *let-60(ga89*gf*)* animals raised at the restrictive temperature by hypoxia. Arrowheads point at the stacked oocytes formed in the proximal gonad under normoxia. **(F)** Suppression of the duct cell duplication phenotype in *let-60(n1046*gf*)* mutants by hypoxia. Arrows point at the duct cell nuclei expressing LIN-48::GFP formed under normoxia (top) and hypoxia (bottom). **(C, D)** Error bars in (C) and (D) indicate the 95% confidence intervals, and *P*-values, indicated with ***$P < 0.001$ and **$P < 0.01$, were derived by bootstrapping 1,000 samples. **(E, F)** In (E) and (F), error bars indicate the standard error of the mean, and *P*-values were calculated with a Fisher's exact test. The numbers of animals scored are indicated in brackets. The scale bars represent 5 μm.

complex and is degraded by the 26S proteasome (Bishop et al, 2004). Under low oxygen concentrations, HIF-1 is stabilized because of decreased EGL-9 activity and forms a complex with the constitutively expressed HIF-β subunit AHA-1 to promote the expression of specific target genes (Bishop et al, 2004; Shen et al, 2005).

We have previously used quantitative genetics to identify modifiers of the RAS/MAPK pathway by comparing two highly polymorphic *C. elegans* strains, N2 Bristol and CB4856 Hawaii, which

had been sensitized by introgression of the G13E *let-60(n1046) ras* gain-of-function allele (Schmid et al, 2015). Notably, CB4856 Hawaii exhibits an overall elevated sensitivity to RAS/MAPK signaling compared with the N2 Bristol reference strain (Milloz et al, 2008; Schmid et al, 2015). Among the RAS/MAPK modifiers identified were *F44F1.1*, which encodes a calpain paralog (*F44F1.3*), and *pfd-3*, a prefoldin orthologous to the human VHL-binding protein 1. Both genes have independently been identified in a screen for suppressors

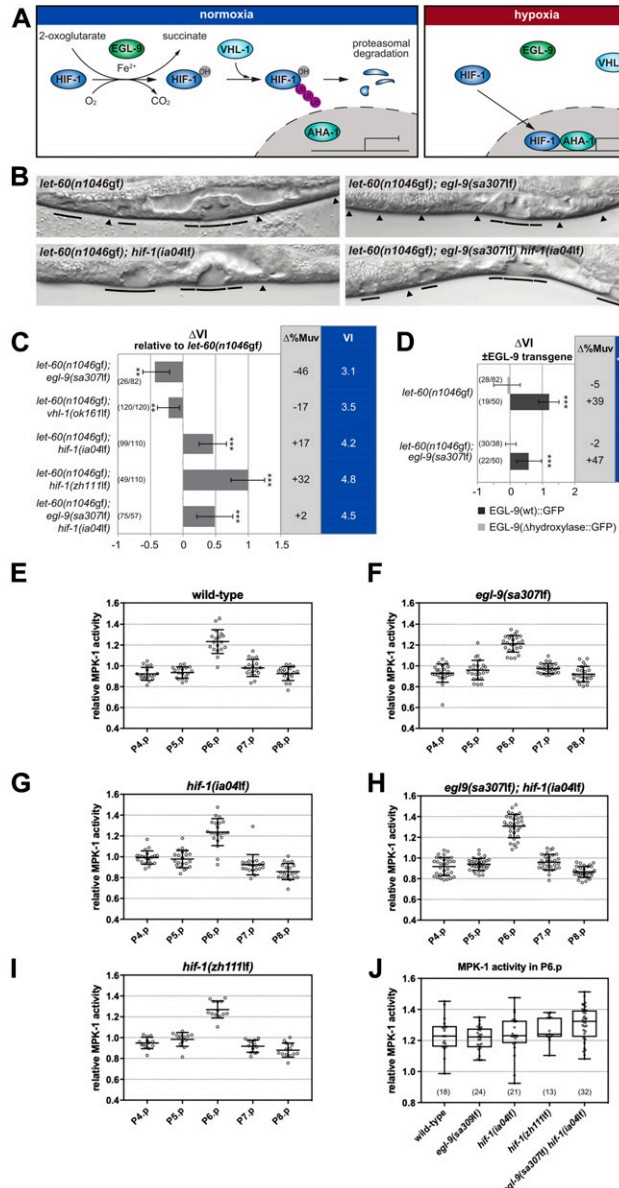

**Figure 2. The hypoxia-response pathway negatively regulates VI under normoxia.**

**(A)** Schematic overview of the conserved hypoxia-response pathway. The *C. elegans* gene names are indicated. **(B)** Vulval phenotypes of double and triple mutants between *let-60(n1046gf)* and components of the hypoxia-response pathway under normoxia. Solid lines indicate induced 1° and 2° and arrowheads uninduced 3° VPCs in the L4 larvae. **(C)** Mutations in the hypoxia-response pathway change the VI of *let-60(n1046gf)* mutants. ΔVI indicates the change in VI of the genotypes relative to *let-60(n1046gf)* single mutant siblings obtained from the crosses. Δ%Muv indicates the change in the percentage of animals with VI > 3. The absolute VIs of the double/triple mutants are shown in the rightmost blue column. **(D)** Overexpression of wild-type *egl-9::gfp* increases the VI. ΔVI indicates the change in VI of animals carrying a wild-type (dark bars) or hydroxylase deficient (light bars) multi-copy *egl-9::gfp* array compared with siblings without array. Error bars indicate the 95% confidence intervals. *P*-values, indicated with ***P < 0.001 and **P < 0.01, were derived by bootstrapping 1,000 samples. **(E–I, J)** MPK-1 biosensor (ERK-nKTR) activity values measured in the VPCs of mid-L2 larvae with the indicated mutant backgrounds and **(J)** comparison of the MPK-1 activity levels in P6.p across the different genotypes. Relative MPK-1 activity values were measured as described under the Materials and Methods section. The numbers of animals scored are indicated in brackets. The scale bar represents 5 μm.

of the *egl-9* prolyl hydroxylase egg-laying–defective phenotype, suggesting that F44F1.3 and *pfd-3* interact with the core hypoxia-response pathway (Gort et al, 2007). Together, these findings suggested that genetic variation in genes controlling the hypoxia-response pathway might account for the differences in RAS/MAPK signaling strength between the N2 Bristol and CB4856 Hawaii *C. elegans* isolates.

Here, we show that hypoxia inhibits the cellular responses to RAS/MAPK signaling in different tissues of *C. elegans*. Surprisingly, only part of the hypoxic inhibition of RAS/MAPK signaling depends on HIF-1 activity. We further identify the nuclear hormone receptor gene *nhr-57* as a critical HIF-1 target that opposes RAS/MAPK–induced differentiation under normoxia and under hypoxia. Finally, we show that the prolyl hydroxylase EGL-9 is a NOTCH target gene. By inducing *egl-9* expression, lateral NOTCH signaling between the proximal VPCs promotes HIF-1 degradation and thereby represses *nhr-57* expression at the onset of vulval fate specification to keep the proximal VPCs competent to respond to the inductive AC signal. This regulatory network allows the adaptation of development to variations in oxygen availability.

## Results

### Hypoxia suppresses LET-60 RAS gain-of-function phenotypes in different tissues

The two polymorphic genes *F44F1.1* and *pfd-3* have previously been found to modify RAS/MAPK signaling during vulval development and to suppress the egg-laying–defective phenotype of *egl-9*(lf) mutants (Gort et al, 2007; Schmid et al, 2015). We, thus, asked if hypoxia modulates RAS/MAPK–mediated cell fate decisions and whether the differences in RAS/MAPK signaling strength observed between different *C. elegans* wild isolates (Milloz et al, 2008) originate from genetic variation in components of the hypoxia-response pathway. To this aim, we introduced by CRISPR/Cas9 genome editing the G13E gain-of-function mutation into the N2 Bristol (*zh121*gf) and CB4856 Hawaii (*zh122*gf) *let-60 ras* locus (Beitel et al, 1990; Arribere et al, 2014). We then determined the VI of N2 *let-60(zh121*gf*)* and CB4856 *let-60(zh122*gf*)* animals raised under different oxygen concentrations. Under normoxia (21% $O_2$), CB4856 animals carrying the G13E mutation exhibited a higher VI than N2 animals carrying the same mutation, as had been observed earlier by introgressing the canonical G13E allele *let-60(n1046*gf*)* into the CB4856 background (Fig 1B and C) (Milloz et al, 2008; Schmid et al, 2015). VI was not significantly changed in either strain when the animals were raised under mild hypoxic conditions of 11%, 7%, or 3% $O_2$ (Fig 1B and C). However, at 1% $O_2$, the VI decreased in the CB4856 background, and at 0.5% $O_2$, the VI was reduced to the same level in both strains. Because 0.5% is the lowest $O_2$ concentration, under which the animals develop to adulthood, we could not test lower $O_2$ concentrations. It, thus, appears that the same baseline level of VI is reached under severe hypoxia (0.5% $O_2$) irrespective of the genetic background.

To further investigate the effect of oxygen on RAS/MAPK signaling and rule out allele-specific effects, we examined the effect of

strong hypoxia (0.5% $O_2$) on different RAS/MAPK pathway mutants in the N2 Bristol background. We included the Wnt pathway–mutant *bar-1(ga80)* and the NOTCH alleles *lin-12(n137gf)* and *lin-12(n137n720lf)* in this analysis. Even though hypoxia had no effect on the VI in the N2 Bristol wild-type strain (VI = 3, n > 50), hypoxia reduced the VI in animals carrying rf mutations in *let-23 egfr*, *let-60 ras*, or *lin-45 raf* or in animals expressing an activated MAPK (MPK-1) from a transgene (*mpk-1(gaIs37gf)*) (Fig 1D). (Note that for each genotype, ΔVI indicates the difference in the induction index between siblings grown at 0.5% and 21% $O_2$.) By contrast, the increased VI in *lin-1(lf)* mutants, which encodes an Ets family transcription factor that represses VI downstream of MPK-1 (Beitel et al, 1995), was not affected by hypoxia. Moreover, the VI of *bar-1 β-catenin* or *lin-12 notch* mutants was not changed by hypoxia, indicating that hypoxia specifically affects the activity of or response to RAS/MAPK signaling (Fig 1D).

We next examined the effect of hypoxia on other tissues, in which RAS/MAPK signaling controls different cell fate decisions. The temperature-sensitive *let-60(ga89gf)* allele causes an accelerated exit of meiotic germ cells from the pachytene stage at the restrictive temperature and results in the stacking of smaller oocytes in the proximal gonad arms (Fig 1E) (Eisenmann & Kim, 1997). Furthermore, the excretory system of *let-60(n1046gf)* mutants frequently contains two duct cells expressing the *lin-48::gfp* marker (Fig 1F) (Berset et al, 2005). Hypoxia partially suppressed the *let-60(gf)* phenotypes in both tissues (Fig 1E and F).

### The canonical hypoxia-response pathway opposes RAS/MAPK–induced VI

Because oxygen serves as an essential energy substrate, a reduction in the atmospheric oxygen concentration induces the adaptation of many metabolic processes (Semenza, 2001). In addition to these immediate metabolic changes, cells activate the hypoxia-response pathway via the hypoxia-inducible factor HIF-1 to change gene expression. Interestingly, VI was significantly reduced only below a threshold oxygen concentration between 1 and 0.5% oxygen rather than decreasing progressively along an $O_2$ gradient (Fig 1C), suggesting the existence of a regulatory switch rather than a systemic metabolic effect of hypoxia on RAS/MAPK signaling. We, thus, examined the genetic interactions between mutations in the RAS/MAPK and hypoxia-response pathways under normoxic conditions (21% $O_2$). For this purpose, we calculated the change in VI (ΔVI) between the indicated double mutants and *let-60(n1046gf)* single mutant control siblings obtained from each cross (Fig 2B and C). The VI of *let-60(n1046gf)* mutants was reduced when combined with the *egl-9(sa307lf)* (Darby et al, 1999) or *vhl-1(ok161lf)* (Epstein et al, 2001) alleles, whereas the VI was increased in combination with the *hif-1(ia04)* allele (Jiang et al, 2001). Because the *ia04* deletion in *hif-1* does not affect all isoforms, we created a larger *hif-1* deletion (*zh111*) that likely represents a null allele (Fig S1A). *let-60(n1046gf); hif-1(zh111lf)* double mutants exhibited an even stronger increase in VI (Figs 2C and S1B). Moreover, *let-60(n1046gf); egl-9(sa307lf) hif-1(ia04lf)* triple mutants exhibited the same increase in VI as *let-60(n1046gf); hif-1(ia04lf)* double mutants, indicating that *hif-1* counteracts RAS/MAPK signaling downstream of *egl-9* prolyl

hydroxylase (Fig 2A–C). *egl-9(sa307lf)* also reduced the VI of mutants in other RAS/MAPK pathway components, such as the guanine nucleotide exchange factor *sos-1* or the *raf* homolog *lin-45* (Fig S2) (Sternberg, 2005).

Because HIF-1 is efficiently degraded under normoxia, the increased VI caused by *hif-1(lf)* mutations under normoxia was somewhat surprising. It, thus, appears that low residual levels of HIF-1 present under normoxia are sufficient to oppose RAS/MAPK signaling. We, therefore, investigated the effect of wild-type and hydroxylase deficient *egl-9::gfp* transgenes on VI (Shao et al, 2009). A wild-type *egl-9::gfp* transgene efficiently rescued the reduced VI of *let-60(n1046gf); egl-9(sa307lf)* and further increased the VI of *let-60(n1046gf)* single mutants (Fig 2D). By contrast, the hydroxylase-deficient *egl-9::gfp* transgene had no effect on the VI in either background. We conclude that EGL-9 enhances RAS/MAPK–induced VPC differentiation by inhibiting HIF-1 protein stability, rather than by acting through a prolyl hydroxylase-independent pathway (Shao et al, 2009; Park et al, 2012). Moreover, EGL-9 enzymatic activity appears to be a rate-limiting step in the inhibition of VI by HIF-1, as the introduction of additional transgenic copies of *egl-9* into an *egl-9(wt)* background caused a further increase in the VI. Thus, low levels of HIF-1 that persist under normoxia are sufficient to repress VI.

### The hypoxia pathway modifies the response to rather than the activity of the RAS/MAPK pathway

Even though mutations in the hypoxia pathway mutants significantly affected VI in sensitized genetic backgrounds, *egl-9(sa307lf)*, *hif-1(ia04lf)*, or *hif-1(zh111lf)* single mutants raised at normoxia or hypoxia exhibited normal vulval development (VI = 3, n > 50 for each condition and genotype). Because VI is very robust even under varying environmental conditions (Félix & Barkoulas, 2012), small alterations in RAS/MAPK pathway activity caused by hypoxia or mutations in the hypoxia-response pathway may not manifest as a vulval fate change, unless the RAS/MAPK pathway is simultaneously compromised. Alternatively, the hypoxia pathway may not directly modify RAS/MAPK pathway activity but rather inhibit the response of the VPCs to RAS/MAPK signaling. To distinguish between these two scenarios, we directly measured MPK-1 activity in the VPCs of hypoxia pathway mutants at the mid-L2 stage that were grown under normoxia. For this purpose, we used a recently developed ERK-nKTR activity biosensor (de la Cova et al, 2017) (for a detailed description of the biosensor quantification, see the Materials and Methods section). All mutant backgrounds exhibited the highest MPK-1 activity levels in P6.p, as observed in wild-type controls (Fig 2E–I). Furthermore, we found no significant decrease in MPK-1 activity in P6.p of *egl-9(sa307)* and no increase in *hif-1(ia04lf)* or *hif-1(zh111lf)* animals relative to the wild-type controls (Fig 2J). (All *P*-values in a one-way ANOVA test with Dunnett's multiple comparison correction were above 0.05.) Thus, we did not detect substantial changes in MPK-1 activity within the dynamic range of the ERK-nKTR biosensor. We conclude that the hypoxia-response pathway attenuates the response of the VPCs to MAPK signaling, rather than directly inhibiting RAS/MAPK pathway activity.

## HIF-1 inhibits VI cell autonomously in the VPCs

Because HIF-1 exerts both cell autonomous and cell non-autonomous functions (Shao et al, 2010; Sendoel et al, 2010), we investigated in which tissue HIF-1 functions to inhibit VI. First, we analyzed the expression pattern of a translational *hif-1::gfp* reporter (Sendoel et al, 2010). The *hif-1::gfp* reporter was undetectable under normoxia, but visible in many somatic cells, including the VPCs and the intestinal cells in *egl-9(sa307*lf*)* or *vhl-1(ok161)* mutants (Fig 3A). Because we had previously described a cell non-autonomous suppression of VI via the *amx-2* monoamine oxidase acting in intestinal cells (Schmid et al, 2015) and because expression of *egl-9::gfp* was prominent in intestinal cells, we tested a possible function of HIF-1 in the intestine and VPCs. To this aim, we performed Pn.p cell–specific and intestine-specific RNA interference using *rde-1(ne219*lf*); let-60(n1046*gf*)* RNAi-resistant mutants expressing *rde-1(wt)* from the Pn.p cell–specific *lin-31* and the intestine-specific *elt-2* promoters (Schmid et al, 2015). Whereas the knockdown of *hif-1* in the intestinal cells did not change the VI, Pn.p cell–specific *hif-1* RNAi significantly increased the VI compared with empty vector control animals (Fig 3B). It is noteworthy that, for unknown reasons, the genetic background used to establish the tissue-specific RNAi strains also affected the VI of the *let-60(n1046*gf*)* mutation, as reported previously (Schmid et al, 2015). Nevertheless, these results indicate that HIF-1 acts autonomously in the VPCs to oppose RAS/MAPK signaling during VI.

## EGL-9 is a NOTCH target gene

Whereas *hif-1::gfp* reporter expression was not detectable in wild-type animals raised under normoxia (Fig 3A), the expression of the functional *egl-9::gfp* reporter could be observed in the VPCs and their descendants throughout vulval development (Fig 3C and D). At the onset of VI in mid L2 larvae, the *egl-9::gfp* reporter was expressed in all VPCs, although expression levels were slightly higher in the three proximal VPCs P5-7.p than the three distal VPCs P3.p, P4.p, and P8.p (Fig 3C, top panel). In late-L2/early-L3 stage animals, EGL-9::GFP expression remained prominent in the proximal VPCs but decreased in the distal VPCs as they started dividing (Fig 3C, second panel). After completion of the first round of VPC divisions in mid-L3 larvae, EGL-9::GFP expression faded in the 1° P6.p descendants and increased in the 2° P5.p and P7.p descendants (Fig 3C, third panel).

This expression pattern suggested that either RAS/MAPK signaling inhibits or lateral DELTA/NOTCH signaling induces EGL-9::GFP expression in the VPCs. We distinguished between these two scenarios by introducing the *egl-9::gfp* reporter into *let-60(n2021*rf*), lin-12(n137n720*lf*)*, and *lin-12 (n137*gf*)* mutants and quantifying the EGL-9::GFP signal intensities. EGL-9::GFP expression was strongly reduced in *let-60(n2021*rf*); egl-9(sa307*lf*)* mutants (Fig 3C and E; note that for unknown reasons we only obtained viable progeny from *let-60(n2021*rf*)* mutants carrying the *egl-9::gfp* transgene in the *egl-9(sa307*lf*)* background). Furthermore, EGL-9::GFP expression was absent in the VPCs of *lin-12(n137n720*lf*)* mutants, whereas *lin-12(n137*gf*)* mutants showed strong and uniform expression in all VPCs (Fig 3C, F, and G). Note that, because no AC is formed in *lin-12(n137*gf*)* mutants, the RAS/MAPK is not activated in *lin-12(n137*gf*)*

mutants. Furthermore, we identified eight predicted CSL (LAG-1)-binding sites (G/ATGGGAA) within a 6-kb *egl-9* genomic region (Fig S3), which is a significant enrichment in CSL-binding site frequency (one site per 8 kb corresponds to a random distribution) and comparable with the known LIN-12 NOTCH target genes *lip-1, lin-12, lag-1*, or *glp-1* (Christensen et al, 1996; Berset et al, 2001; Yoo & Greenwald, 2005). Finally, we tested if *egl-9* modulates lateral NOTCH signaling strength by determining the VI of *lin-12; egl-9(sa307*lf*)* double mutants. The *egl-9(sa307*lf*)* allele neither affected the VI of *lin-12(n137n720*lf*)* nor of *lin-12(n137*gf*)* mutants (Fig S4).

Taken together, we conclude that EGL-9 does not regulate LIN-12 NOTCH signaling but rather acts downstream of LIN-12 NOTCH to down-regulate HIF-1. LIN-12 up-regulates EGL-9::GFP expression independently of RAS/MAPK signaling, and the enrichment of CSL-binding sites in the *egl-9* locus suggests that *egl-9* may be a direct LIN-12 NOTCH target gene. Because RAS/MAPK signaling induces the expression of the DSL NOTCH ligands (Chen & Greenwald, 2004) (Fig 1A), inhibiting RAS/MAPK signaling most likely reduces EGL-9::GFP expression indirectly via the loss of NOTCH activation.

## The HIF-1 target gene *nhr-57* inhibits VI under normoxia

HIF-1 up-regulates the expression of numerous target genes under hypoxia and induces multiple cellular responses (Semenza, 2001). Previous studies have identified a large number of HIF-1 regulated genes in *C. elegans* (Bishop et al, 2004; Shen et al, 2005). We used the available expression data to screen for genes that act downstream of HIF-1 and negatively regulate VI. We selected a total of 80 predicted HIF-1 target genes that exhibited an at least twofold induction by HIF-1 and performed RNAi in *let-60(n1046*gf*); egl-9(sa307*lf*)* and *let-60(n1046*gf*); hif-1(zh111*lf*)* double mutants grown under normoxia (Tables S1 and S2). We used these two double-mutant backgrounds to distinguish between RAS/MAPK pathway modifiers that are partially or fully HIF-1 dependent. We hypothesized that the knockdown of a gene that inhibits VI and depends on HIF-1 for its expression should increase the VI of *let-60(n1046*gf*); egl-9(sa307*lf*)* mutants in which HIF-1 is stabilized. However, knockdown of the same gene in *let-60(n1046*gf*); hif-1(zh111*lf*)* mutants should not increase the VI because this gene would not be expressed in the absence of HIF-1.

Based on these criteria, we identified a single HIF-1 target gene, *nhr-57*, as a negative regulator of RAS/MAPK-induced VPC differentiation (Fig 4A and Table S1). *nhr-57* encodes a nuclear hormone receptor similar to the *Drosophila* estrogen-related receptor ERR2 and the vertebrate glucocorticoid hormone receptor NR3C1 (Shaye & Greenwald, 2011). We next used the *nhr-57(tm4533*lf*)* deletion allele to confirm that NHR-57 acts as an inhibitor of VPC differentiation. The VI of *let-60(n1046*gf*); hif-1(zh111*lf*); nhr-57(tm4533*lf*)* triple mutants was not significantly different from *let-60(n1046*gf*); hif-1(zh111*lf*); nhr-57(+)* control siblings (Fig 4B). However, the *nhr-57(tm4533*lf*)* allele increased the VI of *let-60(n1046*gf*); egl-9(sa307*lf*)* double and of *let-60(n1046*gf*)* single mutants compared with the corresponding *nhr-57(+)* control siblings.

We next assessed the expression pattern of NHR-57 by constructing a translational *nhr-57::gfp* reporter transgene. The reporter includes the complete *nhr-57* promoter/enhancer region and contains the predicted HIF-1–binding sites (Miyabayashi et al,

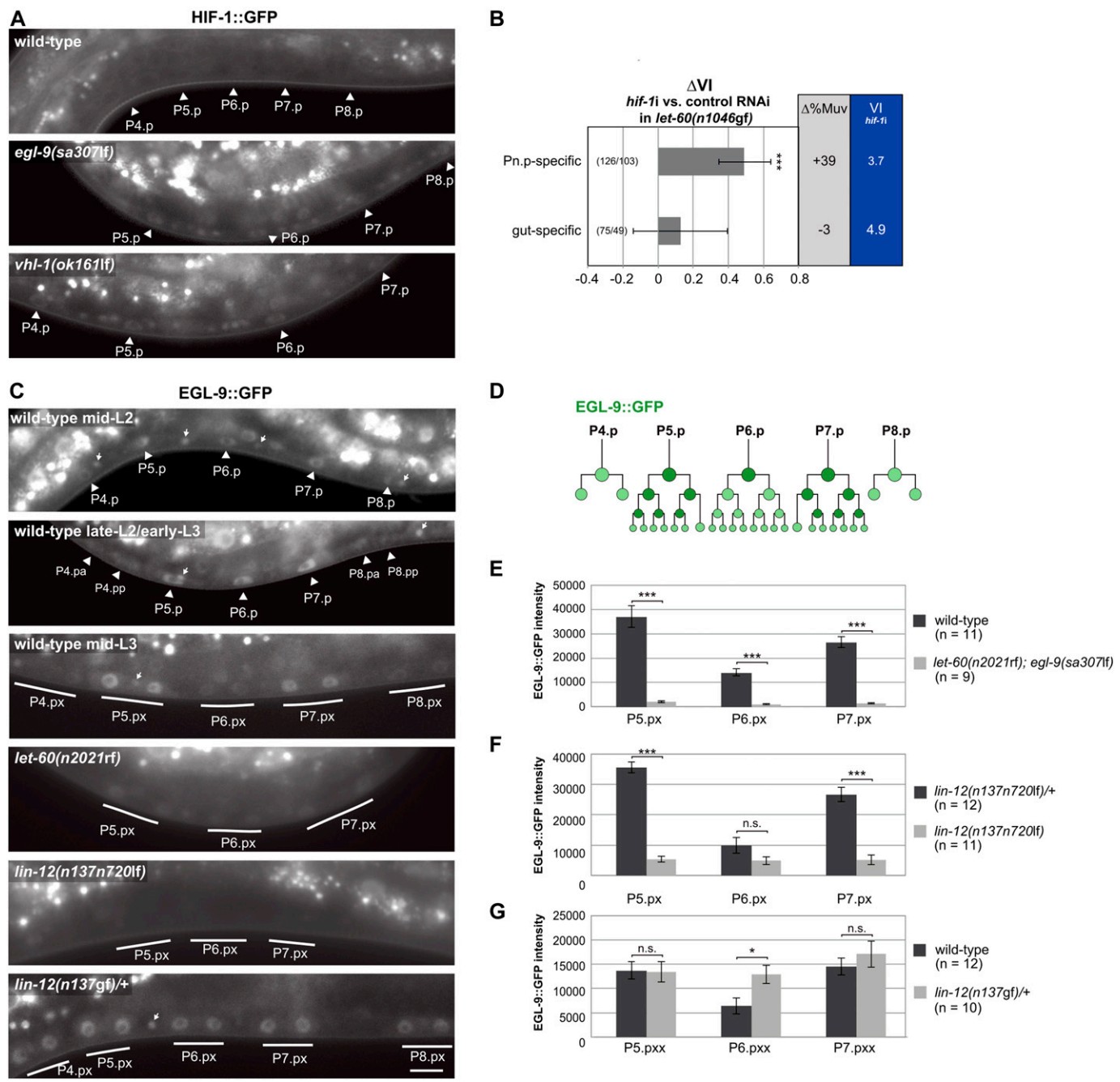

**Figure 3. *egl-9* is a NOTCH target that promotes HIF-1 degradation in the induced VPCs.**
**(A)** HIF-1::GFP expression in wild-type, *egl-9(sa307*lf*)*, and *vhl-1(ok161*lf*)* larvae before VI. Arrowheads point at the nuclei of the VPCs. **(B)** Pn.p cell– and gut-specific *hif-1* RNAi (see main text and the Material and Methods section). ΔVI indicates the change in VI in *hif-1* RNAi compared with empty vector–treated control animals. Error bars indicate the 95% confidence interval, and *P*-values were derived by bootstrapping 1,000 samples. **(C)** EGL-9::GFP expression in a wild-type mid-L2 larva (top panel) at the onset of VI, a late-L2/early-L3 larva before the proximal and after the distal VPCs had divided (second panel) and a mid-L3 larva after all VPCs had divided once (third panel, Pn.px stage). The arrowheads point at the VPC nuclei and the small arrows at the nuclei of ventral nerve cord neurons expressing EGL-9::GFP. Expression in *let-60 ras(n2021*rf*)*, *lin-12 notch(n137n720*lf*)*, and *lin-12 notch(n137*gf*)* mutants is shown in mid-L3 larvae, after the first round of VPC divisions had been completed (Pn.px stage). Induced VPCs are underlined. The scale bar represents 5 μm. **(D)** Schematic representation of the wild-type EGL-9::GFP pattern during vulval development. **(E–G)** Quantification of EGL-9::GFP expression in the four genotypes shown in (C) after VI (early to mid L3 and at the Pn.pxx stage in (G)). Error bars indicate the standard error of the mean, and *P*-values, indicated with ***$P < 0.001$ and *$P < 0.05$, were calculated in a two-tailed *t* test. The numbers of animals scored are indicated in brackets.

1999; Shen et al, 2006). In animals raised under normoxia, the *nhr-57::gfp* reporter was weakly expressed in several somatic cells including the intestinal cells, but not in the VPCs before VI. After the

first round of VPC divisions in mid-L3 larvae, *nhr-57::gfp* expression became visible in the descendants of the uninduced distal VPCs but remained undetectable in the descendants of the induced proximal

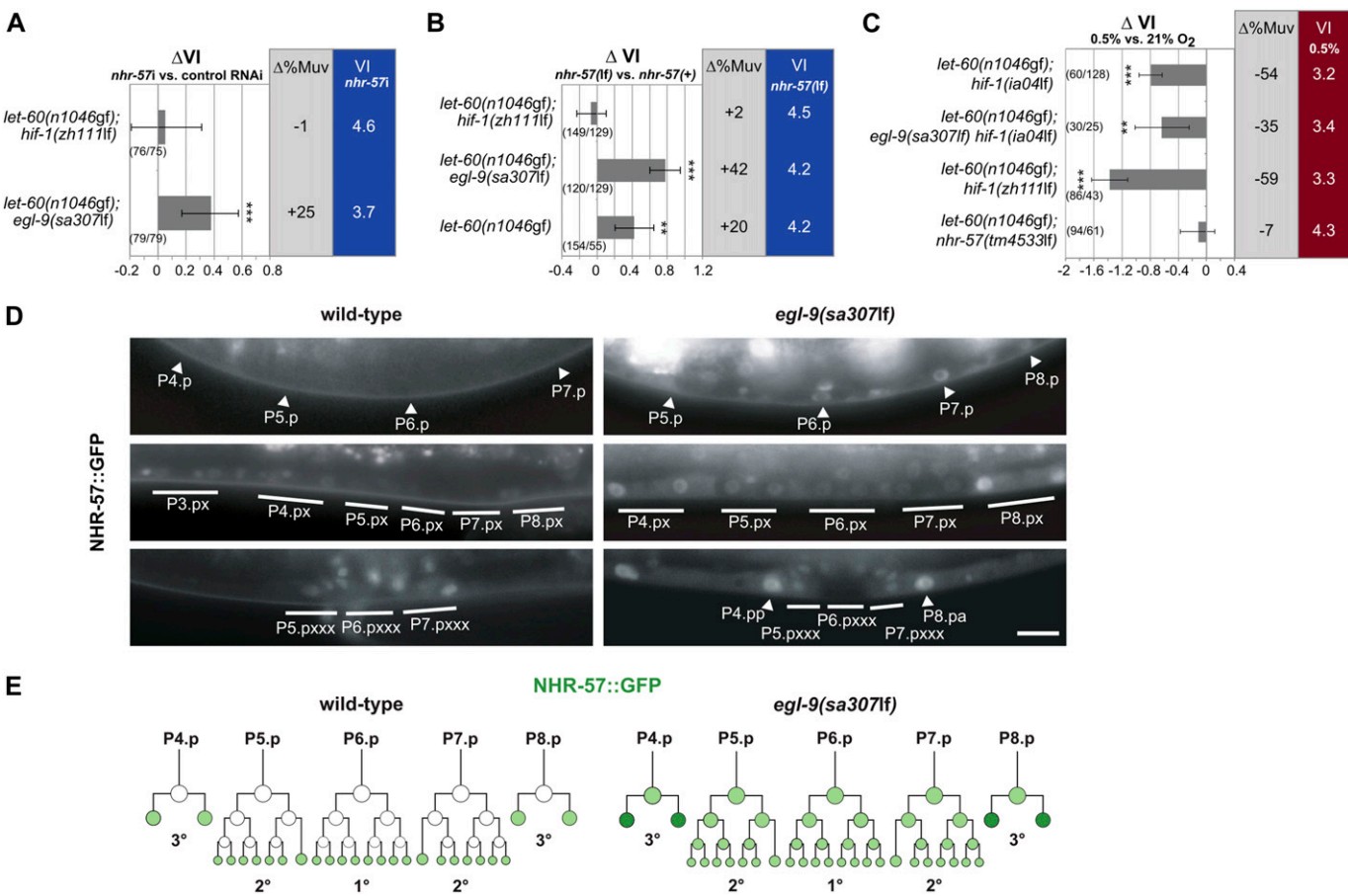

**Figure 4. The HIF-1 target NHR-57 inhibits RAS/MAPK signaling under normoxia and hypoxia.**
**(A)** Knock-down of *nhr-57* increases the VI of *let-60(n1046gf)* in an *egl-9(sa307*lf*)* but not in a *hif-1(zh111*lf*)* background. ΔVI and Δ%Muv indicate the changes in VI and percentage of animals with VI > 3 after *nhr-57* RNAi compared with empty vector–treated control animals. **(B)** Deletion of the *nhr-57(tm4533*lf*)* allele increases the VI of *let-60(n1046gf)* single and *let-60(n1046gf); egl-9(sa307*lf*)* double mutants but has no effect in a *let-60(n1046gf); hif-1(zh111*lf*)* background. ΔVI and Δ%Muv indicate the changes in VI and the percentage of animals with VI > 3 compared with *nhr-57(+)* control siblings. **(C)** Hypoxic treatment decreases the VI in the *hif-1(ia04*lf*)*, *hif-1(zh111*lf*)*, and *egl-9(sa307*lf*) hif-1(ia04*lf*)* but not the *nhr-57(tm4533*lf*)* background. ΔVI and Δ%Muv indicate the changes in VI and percentage of animals with VI > 3 raised in hypoxia compared with normoxia. Error bars indicate 95% confidence intervals, and *P*-values, indicated with ***P < 0.001 and **P < 0.01, were derived by bootstrapping 1,000 samples. The numbers of animals scored are indicated in brackets. **(D)** NHR-57::GFP expression pattern in the wild-type (left panels) and *egl-9(sa307*lf*)* (right panels) background in late-L2/early-L3 larvae at the Pn.p stage (top panels), in mid L3 larvae (Pn.px stage) after induction (middle panels), and at the end of vulval differentiation (bottom panels). Arrowheads and solid lines point at the nuclei of the induced or uninduced VPCs and their descendants, respectively. The scale bar represents 5 *µ*m. **(E)** Schematic representation of the NHR-57::GFP pattern during vulval development in the wild-type (left) and in an *egl-9(sa307*lf*)* background (right).

VPCs until they had completed their last round of division and terminally differentiated (Fig 4D and E). Consistent with its function as an HIF-1 target, NHR-57::GFP expression was up-regulated in the VPCs of *egl-9(sa307*lf*)* mutants with the signal being most prominent in the descendants of uninduced VPCs (Fig 4D and E). Taken together, we propose that LIN-12 NOTCH signaling induces EGL-9 expression in the proximal VPCs to counteract the inhibitory effect of HIF-1 and its target NHR-57 on VPC fate specification.

### NHR-57 mediates the HIF-1–dependent and HIF-1–independent hypoxic inhibition of VI

Our results so far have suggested that under normoxic conditions, EGL-9 induces HIF-1 degradation to prevent the HIF-1 target NHR-57 from inhibiting vulval differentiation (Fig 4A and B). However,

HIF-1–independent factors also play a role in the transcriptional response to hypoxia (Shen et al, 2005; Padmanabha et al, 2015). We, thus, investigated if the inhibitory effect of hypoxia on vulval development depends entirely on HIF-1 activity. Surprisingly, the VI of *let-60(n1046gf); hif-1(*lf*)* double and *let-60(n1046gf); egl-9(sa307*lf*) hif-1(ia04*lf*)* triple mutants was significantly decreased under hypoxia (Fig 4C). Therefore, the reduction in VI under hypoxia only partially depends on HIF-1.

NHR-57 appeared to be a good candidate for a factor that mediates the HIF-1–independent effect of hypoxia for a number of reasons. First, the translational *nhr-57::gfp* reporter was expressed under normoxia in cells that also expressed the *egl-9::gfp* reporter. Second, *nhr-57::gfp* expression in the VPCs of *egl-9(sa307*lf*)* mutants that uniformly express *hif-1::gfp* (Fig 3A) was not uniform (Fig 4D), pointing at a regulation of *nhr-57* expression by additional factors

besides HIF-1. Third, the transcription of *nhr-57* was significantly enhanced under hypoxia even in the absence of HIF-1 activity (Bishop et al, 2004). Together, these findings suggested that NHR-57 integrates multiple inputs together with the HIF-1–mediated hypoxic response. In contrast to *let-60(n1046*gf*); hif-1*(lf*)* mutants, the VI of *let-60(n1046*gf*); nhr-57(tm4533*lf*)* double mutants did not change under hypoxia (Fig 4C). Thus, NHR-57 is necessary for both the HIF-1–dependent and HIF-1–independent repression of VI by hypoxia.

# Discussion

We have found that different tissues in *C. elegans* can modulate their cellular responses to RAS/MAPK signaling according to the availability of oxygen. Under limiting oxygen concentrations, energetically demanding processes that are dispensable for viability, such as oogenesis or VI, may be suppressed to preserve essential cellular functions. By connecting the cellular hypoxia response to an essential developmental signaling pathway, the animals can adapt their development to fluctuations in oxygen concentration. In wild-type animals raised under the optimal laboratory growth conditions, the effect of hypoxia does not manifest as a change in cell fate specification in the tissues we examined. The effect of hypoxia only becomes apparent if the RAS/MAPK pathway has been sensitized. However, the hypoxic effect may become relevant when animals in the wild encounter stressful conditions, in which developmental robustness is challenged. The capability to adapt to varying oxygen concentrations could increase developmental robustness.

The CB4856 Hawaii wild isolate exhibits overall higher levels of VI than the N2 Bristol reference laboratory strain when grown under normoxia (Milloz et al, 2008; Schmid et al, 2015) as well as under mild hypoxic conditions (>3% $O_2$) (this study). Strong hypoxia (<1% $O_2$) not only reduces vulval differentiation to a baseline level but also eliminates the difference between the two strains. Thus, differences in their hypoxia response may account for some of the differences in VI between the Bristol N2 and Hawaii CB4856 strains. Most wild *C. elegans* isolates, including CB4856, are adapted to lower oxygen levels and aggregate when grown under normoxia (Hodgkin & Doniach, 1997; Rogers et al, 2006). Aggregation is often followed by burrowing, which may further decrease the oxygen concentrations the animals experience. By contrast, the laboratory strain N2 Bristol does not aggregate because it has become habituated to atmospheric oxygen concentrations during the prolonged laboratory cultivation (de Bono & Bargmann, 1998). Thus, in wild strains, the cellular sensitivity to RAS/MAPK signaling may be elevated to allow development under lower oxygen levels.

We next focused on the process of vulval fate specification to dissect how the hypoxia-response and RAS/MAPK signaling pathways interact at the molecular level. Because the activity of the prolyl hydroxylase EGL-9 is a rate-limiting step in the HIF-1 degradation pathway (Fig 2D), the hypoxia-response pathway exerts its inhibitory effect even under normoxia. This observation allowed us to investigate the cross-talk between the hypoxia and RAS/MAPK pathways under standard growth conditions. By screening a selection of candidate genes regulated by HIF-1, we have identified

the nuclear hormone receptor NHR-57 as a key downstream integrator of both the HIF-1–dependent and HIF-1–independent pathways (Fig 5). Because we did not detect significant changes in MAPK biosensor activity in *hif-1*(lf*)* or *egl-9*(lf*)* mutants, the hypoxia-response pathway and NHR-57 must act downstream of or in parallel with the RAS/MAPK pathway. NHR-57 may interact with the ETS family transcription factor LIN-1, which is an MAPK substrate that represses VPC differentiation when the RAS/MAPK pathway is inactive (Beitel et al, 1995; Jacobs et al, 1998; Tan et al, 1998). Consistent with this hypothesis, multiple genetic and physical interactions between ETS family transcription factors and steroid hormone receptors have been reported in other systems (Mullick et al, 2001; Geng & Vedeckis, 2005; Kalet et al, 2013; Cao et al, 2015). Among the steroid hormone receptors, the glucocorticoid receptor NR3C1 exhibits closest homology to NHR-57 (Shaye & Greenwald, 2011). Also, NR3C1 accumulates under hypoxia to promote erythropoiesis (Bauer et al, 1999; Mense, 2006). Although NHR-57 is a critical HIF-1 target that counteracts RAS/MAPK signaling in the VPCs, NHR-57 expression appears to be regulated by additional factors besides HIF-1 (Bishop et al, 2004; Padmanabha et al, 2015). Similarly, the expression of the mammalian glucocorticoid receptor NR3C1 is not only regulated by hypoxia but also by serotonin (5-HT) (Mitchell et al, 1990). It thus seems likely that *nhr-57* expression is controlled by several additional cues besides the HIF-1–induced up-regulation under hypoxia.

Interestingly, the regulation of *egl-9* expression by NOTCH signaling provides a link between the VPC fate specification and the hypoxia-response pathways. This link could be used to modulate the sensitivity of the VPCs to the inductive signal according to oxygen concentrations. At the beginning of VI, EGFR/RAS/MAPK signaling is predominantly activated in the proximal VPCs (P5.p, P6.p, and P7.p) by the LIN-3 EGF signal secreted from the AC. EGFR/RAS/MAPK signaling then up-regulates the expression of multiple DELTA family ligands in P6.p to activate lateral NOTCH signaling in the neighboring VPCs P5.p and P7.p and focus the inductive signal on P6.p, the future 1° VPC (Fig 1A) (Berset et al, 2001; Yoo et al, 2004; Sternberg, 2005). However, in early L3 larvae, both the NOTCH ligand LAG-2 and its receptor LIN-12 are expressed in all VPCs (Chen & Greenwald, 2004; Levitan & Greenwald, 1998). Only after VI before the VPCs divide at the mid-L3 stage, LAG-2 expression becomes restricted to P6.p, whereas LIN-12 levels decrease in P6.p. Thus, reciprocal LIN-12 NOTCH signaling between the VPCs during an early phase of vulval fate specification may induce EGL-9 expression to reduce HIF-1 and NHR-57 levels and keep the proximal VPCs competent to differentiate (Fig 5A). In line with this model, NOTCH signaling has previously been shown to maintain the VPCs competent to respond to the inductive LIN-3 EGF signal even after the VPCs have started dividing (Wang & Sternberg, 1999; Nusser-Stein et al, 2012). The lack of inductive EGFR/RAS/MAPK and lateral NOTCH signaling in the distal uninduced VPCs and their descendants results in higher NHR-57 levels, which reduce their capability to differentiate. However, under hypoxic conditions, the elevated NHR-57 levels may render all VPCs less responsive to the inductive LIN-3 EGF signal (Fig 5B).

The activity of the NOTCH signaling pathway and short-term hypoxic periods involving activation of the hypoxia-response pathway are important during normal animal development

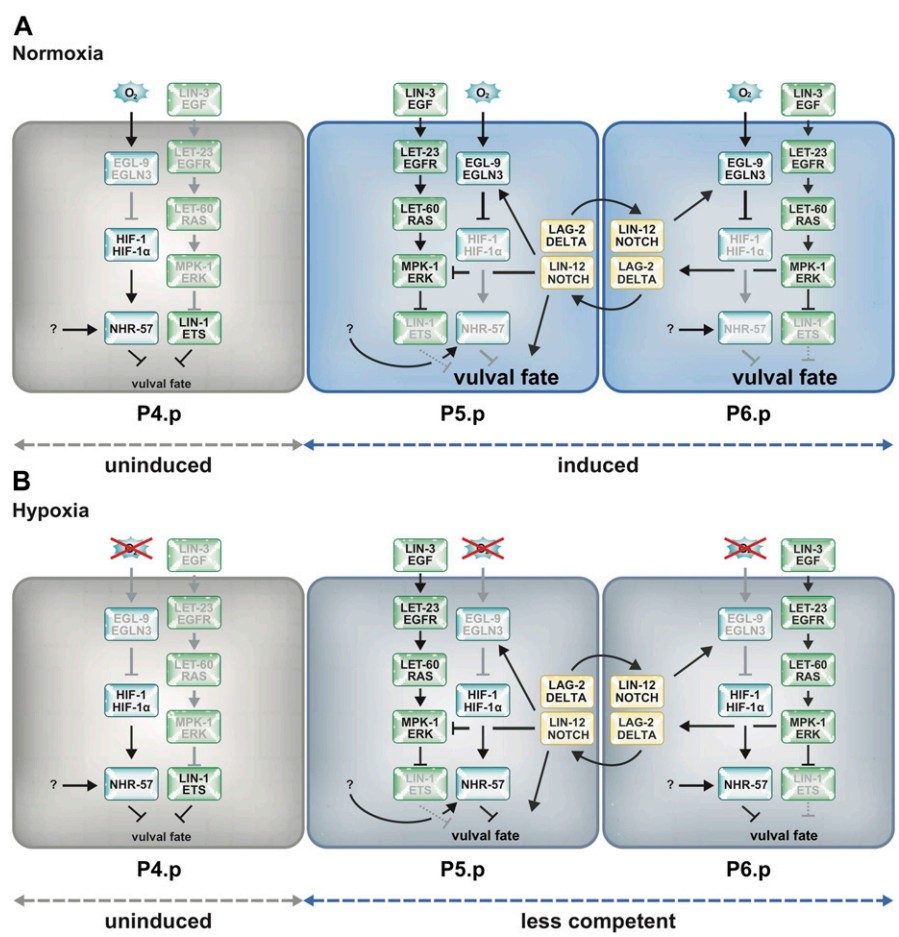

**A**
**Normoxia**

**B**
**Hypoxia**

**Figure 5. EGL-9 inhibits the HIF-1 target NHR-57 from reducing RAS/MAPK signaling through a DELTA/NOTCH-induced negative feedback loop.**
**(A)** Model illustrating the cross-talk of the hypoxia-response, DELTA/NOTCH, and RAS/MAPK pathways at normoxia. At the onset of VI, before a distinction between 1° and 2° VPCs is made, lateral DELTA/NOTCH signaling between the proximal VPCs induces EGL-9 expression to maintain low HIF-1 and NHR-57 levels, thereby keeping the VPCs competent to respond to RAS/MAPK signaling. RAS/MAPK signaling, in turn, promotes the expression of the DELTA family NOTCH ligands. Distal VPCs lose their competence because of higher NHR-57 levels and adopt the 3° fate. Thereafter, NOTCH signaling directly induces the 2° fate and inhibits RAS/MAPK signaling in P5.p and P7.p. **(B)** Under hypoxia, the inhibition of HIF-1 and NHR-57 by EGL-9 is reduced because of the lack of oxygen and vulval fate acquisition is compromised. An as-of-yet unidentified factor activates NHR-57 during hypoxia independently of HIF-1.

(Patterson & Zhang, 2010; Siebel & Lendahl, 2017). To our knowledge, prolyl hydroxylases have so far not been implicated in RAS/MAPK or DELTA/NOTCH–mediated cell fate decisions in other animals. However, a dependence of mammalian EGLN3 hypoxia-inducible factor prolyl hydroxylase transcription downstream of NOTCH has been reported (Ohashi et al, 2011; Li et al, 2012). It remains to be investigated whether these two signaling pathways are inter-connected in other organisms in a similar manner as in *C. elegans* to couple cell fate decision with oxygen availability.

## Materials and Methods

### *C. elegans* alleles

*C. elegans* strains were maintained at 20°C using standard procedures (Brenner, 1974) except N2 strains carrying the *let-60(ga89)* allele that were maintained at 20°C and shifted to 25°C for mutant analysis. *C. elegans* Bristol refers to the wild-type N2 strain and Hawaii to CB4856 (Erik C Andersen, 2015). The transgenic lines were generated as described in (Mello et al, 1991). The following mutations and transgenes were used in this study: **LGII:** *let-23(sy1)*; **LG III:** *unc-32(e189) lin-12(n137n720), dpy-19(e1259) lin-12(n137),*

*hT2[bli-4(e937) let-?(q782) qIs48]* (I;III). **LG IV**: *let-60(n1046), let-60(zh121)* (this study), *let-60(zh122)* (this study), *let-60(ga89), let-60(n2021), lin-1(n304), lin-3(e1417), lin-45(sy96)*. **LG V**: *egl-9(sa307), hif-1(ia04), hif-1(zh111)* (this study), *nhr-57(tm4533)* (Mitani lab). **LG X**: *vhl-1(ok161), rde-1(ne219), bar-1(ga80)*. **Transgenes**: *gaIs37 [lin-31::mpk-1(gf); lin-31::D-mek(gf)]* (Lackner & Kim, 1998), *iaIs38[egl-9p::egl-9::tag, unc-119(+)]* (Shao et al, 2009), *iaEx101[egl-9p::egl-9(H487A)::tag, unc-119(+)]* (Shao et al, 2009), *duIs?[elt-2p::rde-1(+); pRF4]*, *zhEx418[lin-31p::rde-1;myo-2::mCherry]*, *saIs14[lin-48p::gfp]*, *opIs206[hif-1p::hif-1::gfp::hif-1* 3'UTR, *unc-119(+)]* (Sendoel et al, 2010), *zhEx605[nhr-57p::nhr-57::gfp:: nhr-57* 3'UTR, *unc-119(+)]* (this study), *arTi85[lin-31p::ERK-KTR(NLS3)-mClover-T2A-mCherry-H2B::unc-54* 3'UTR, *rps-27p::NeoR::unc-54* 3'UTR] (de la Cova et al, 2017).

Detailed information on the alleles used can be found under www.wormbase.org.

A list of all strains used can be found in Table S3.

### RNAi experiments

RNAi was performed as described in (Kamath et al, 2003). Larvae were synchronized by hypochloride treatment of gravid adults and raised on bacteria expressing dsRNA. The empty vector plasmid L4440 was used as a negative control and dsRNA against *rpn-6.1* as a positive control. VI was scored in the F1 generation. Animals were

anesthetized in in a drop of 5 mM tetramisole and analyzed using differential interference contrast and fluorescence microscopy.

## Staging of the animals used for microscopy analysis

Synchronized population of L1 larvae were obtained by hypochloride treatment of gravid adults and letting the embryos hatch in S-basal buffer without OP50 bacteria for 16–20 h. Arrested L1 larvae were then fed with OP50 bacteria until they reached the stage to be analyzed. The developmental stages were further distinguished using Nomarski image stacks taken along with the fluorescent images by scoring VPC divisions and the dorsal turning of the distal tip cell (DTC) and by measuring the gonad length as described (Kimble & Hirsh, 1979). Developmental stages were assigned as follows: mid-L2 stage (gonad: 35–75 $\mu$m, VPCs undivided, and DTC not turned), late-L2 stage (gonad: 75–110 $\mu$m and DTC not turned), early-L3 stage (gonad: >110 $\mu$m and DTC begins to turn), and mid-L3 stage (gonad: >110 $\mu$m, all VPCs divided once, and DTC turned).

## VI counts

VI was scored by examining worms at the L4 stage under Nomarski optics as described in (Sternberg & Horvitz, 1986). The number of VPCs that had adopted a 1° or 2° vulval cell fate was counted for each animal, and the VI index was calculated by dividing the total number of induced cells by the number of animals scored. Animals with a VI > 3 were scored as hyperinduced and animals with VI < 3 as hypoinduced. The different allele combinations compared were generated from progeny obtained in the same crosses. The raw data used to compile the induction graphs shown in the figures can be found in Table S4.

## ERK-nKTR biosensor quantification

MPK-1 activity in the VPCs was measured in mid-L2 larvae (Pn.p stage) using the recently established ERK-nKTR biosensor (*arTi85*), which is based on the MPK-1 activity-dependent nuclear export of the biosensor (de la Cova et al, 2017). Images were recorded on a Leica widefield microscope equipped with a piezo focus drive, a beam splitter, and two scientific Complementary metal–oxide–semiconductor cameras to simultaneously record the mCherry and mClover signals. We used custom-made ImageJ and CellProfiler scripts provided in the Supplemental Information to process and quantify the images taken in the indicated mutant backgrounds under standardized illumination conditions. Flat field illumination and background corrections were done using blank and dark-field images obtained for every experiment. The nuclear RED/GREEN (mCherry::H2B/nKTR::mClover) average intensity ratios were measured in each VPC (except for P3.p) in summed z-projections of the five central slices (z-spacing of 0.2 $\mu$m) relative to the focus of the nuclear mCherry::H2B signal, using a modified Cell Profiler script as described by de la Cova et al (2017). For each VPC, the RED/GREEN nuclear ratio was normalized to the mean of the RED/GREEN ratios in all VPCs in the same animal, and these values are shown in the graphs of Fig 2. Statistical analysis was carried out with one-way ANOVA followed by Dunnett's multiple comparison correction.

## Hypoxic treatment

Hypoxic chambers were adapted from Fawcett et al (2012) as follows. Briefly, a glass bowl with flange served as container for small Nematode Growth Medium plates. A customized lid was made, which consists of an acrylic glass plate surrounded by a metal ring equipped with an indentation to fit in an O-ring. Two holes in the acrylic glass served as the gas in- and output apertures. A third larger hole accommodates an oxygen measuring device (GOX 100). A hypoxic environment was generated via oxygen replacement by $N_2$ until the desired $O_2$ concentration was obtained. Nematode Growth Medium plates containing mixed-stage *C. elegans* were incubated for 3 to 5 d and analyzed at the microscope.

## Generation of the *hif-1(zh111)* null allele

CRISPR/CAS9 was performed as described in Friedland et al (2013), Arribere et al (2014). 25 ng/$\mu$l *dpy-10* sgRNA pJA58 was co-injected into N2 Bristol animals with 25 ng/$\mu$l of a sgRNA targeting *hif-1* (GATA-GAAAAGTGAGTCCTAA), 500 nM of *dpy-10(cn64)* repair template AF-ZF-827, and 50 ng/$\mu$l of pDD162. Candidate worms were selected from plates harboring a large number of animals showing a roller phenotype progeny and analyzed by PCR. The *zh111* deletion spans 2,464 bp and removes three to five exons (region upstream: ATTT-CAAAAAATTTTTGACA; region downstream: CTAAGTTAAAAAAACAACAG).

## Generation of *let-60(zh121)* and *let-60(zh122)*

The sgRNA sequence AATGACGGAGTACAAGCTTG was inserted into plasmid #46169 (PU6::unc-119_sgRNA) via site-directed mutagenesis. The *let-60* genomic locus was partially amplified using the primers AAGGAGCAAATCGAACAGAC and ATCCATTTTATTAGGCACGCAC and cloned into p-GEM-T easy. The silently mutated PAM and the G13E mutation were obtained using the primers TGAGTGCTGATTTACCAACTCCTC-CATCTCCAACTACCACAAGCTTGTACTCCGTC and AAGCTTGTGGTAGTTGGA-GATGGAGGAGTTGGTAAATCAGCACTCACCATTCAACTCATC. N2 Bristol or CB4856 Hawaii animals were injected with 50 ng/$\mu$l sgRNA plasmid, 50 ng/$\mu$l pDD162, 50 ng/$\mu$l repair template, and 5 ng/$\mu$l pCFJ104. The progeny was screened for the multivulva phenotype, and candidate animals were verified by PCR and sequencing.

## Translational *nhr-57* reporter

A 1,206-bp fragment upstream of the *nhr-57* transcriptional start site was amplified together with the coding genomic region (without the stop codon) using the primers ctaacaacttggaaatgaaatCACCAA-CACCTTCTACACAGCTGC and gttcttctcctttactcatTTGTCCATCAA-TGATTTTATAGATTTTGTCG. The *gfp* sequence was amplified from pPD95.75 using the primers ATGAGTAAAGGAGAAGAAC and gagatctggttcaaatagCTATTTGTATAGTTCATCCATGC. 459 bp of the *nhr-57* 3′ UTR were amplified using the primers GCTATTTGAACCAGATCTCTTC and cagtacggccgactagtagGAATAAATCATCCCAAAGCCGTTTTTG. A vector backbone containing a *CB-unc-119(+)* rescue as well as the *AmpR* gene was amplified using the primers ATTTCATTTCCAAGTTGTTAGCG and CTACTAGTCGGCCGTACTGAGGTGTTGTCGCTT-TTATTGGG. The four fragments were combined into a 10.170-kb plasmid (pSMa32) using Gibson Assembly. N2 animals were injected (because of mutation in

the *CB-unc-119(+)* gene) with 5 ng/μl pSMa32, 2.5 ng/μl pCFJ104 and 100 ng/μl pBS. Transformants were isolated based on the presence of the co-injection marker.

## Supplementary Information

## Acknowledgements

We wish to thank the members of the Hajnal laboratory, Jan Kammenga, Beatrice Beck-Schimmer, and Konrad Basler for critical discussion and comments on the manuscript, and R Maier and D Schnarwiler for their help in designing and manufacturing the hypoxic chambers. We are also grateful to the *C. elegans* Genetics Center CGC, which is funded by NIH Office of Research Infrastructure Programs (P40 OD010440), and the Mitani lab (National Bioresource Project) for providing some strains, Andrew Fire for GFP vectors, and J Ahringer for RNAi clones. This work was supported by a grant from the Swiss National Science Foundation to A.H. no. 31003A-166580 and the Kanton of Zürich.

### Author Contributions

S Maxeiner: conceptualization, data curation, formal analysis, supervision, investigation, and writing—original draft.
J Grolleman: validation and investigation.
T Schmid: conceptualization, resources, data curation, and investigation.
J Kammenga: funding acquisition and project administration.
A Hajnal: conceptualization, data curation, formal analysis, supervision, funding acquisition, investigation, project administration, and writing—review and editing.

### Conflict of Interest Statement

The authors declare that they have no conflict of interest.

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
