## [Reviewer comments · Life Science Alliance]

Life Science Alliance

The hypoxia-response pathway modulates RAS/MAPK-mediated cell fate decisions in *C. elegans*

Sabrina Maxeiner, Judith Grolleman, Tobias Schmid, Jan Kammenga, and Alex Hajnal

DOI: <https://doi.org/10.26508/lsa.201800255>

Corresponding author(s): Alex Hajnal, University of Zurich

Review Timeline:	Submission Date:	2018-11-26
	Editorial Decision:	2019-01-10
	Revision Received:	2019-04-16
	Editorial Decision:	2019-05-13
	Revision Received:	2019-05-16
	Accepted:	2019-05-17

Scientific Editor: Andrea Leibfried

Transaction Report:

January 10, 2019

Re: Life Science Alliance manuscript #LSA-2018-00255-T

Dr. Alex Hajnal
University of Zurich
Institute of Molecular Life Sciences
Winterthurerstr. 190
Zurich CH-8057
Switzerland

Dear Dr. Hajnal,

Thank you for submitting your manuscript entitled "Cross-talk between the NOTCH and hypoxia-response pathways modulates RAS/MAPK-mediated cell fate decisions in *C. elegans*" to Life Science Alliance. The manuscript was assessed by expert reviewers, whose comments are appended to this letter.

As you will see, the reviewers appreciate your data. However, while reviewer #1 and #2 support publication of your work upon minor revision, reviewer #3 is concerned by how the *C. elegans* staging and VPC fate analysis was performed and by the statistical analysis conducted, questioning the validity of your conclusions. We would thus like to invite you to revise your work to address the minor concerns of reviewer #1 and #2 by text changes and, importantly, the major concerns of reviewer #3. We would be happy to discuss the individual revision points further with you should this be helpful.

Thank you for this interesting contribution to Life Science Alliance. We are looking forward to receiving your revised manuscript.

Sincerely,

- A letter addressing the reviewers' comments point by point.
- An editable version of the final text (.DOC or .DOCX) is needed for copyediting (no PDFs).
- High-resolution figure, supplementary figure and video files uploaded as individual files: See our detailed guidelines for preparing your production-ready images, <http://life-science-alliance.org/authorguide>
- Summary blurb (enter in submission system): A short text summarizing in a single sentence the study (max. 200 characters including spaces). This text is used in conjunction with the titles of papers, hence should be informative and complementary to the title and running title. It should describe the context and significance of the findings for a general readership; it should be written in the present tense and refer to the work in the third person. Author names should not be mentioned.

B. MANUSCRIPT ORGANIZATION AND FORMATTING:

Full guidelines are available on our Instructions for Authors page, <http://life-science-alliance.org/authorguide>

Reviewer #1 (Comments to the Authors (Required)):

The main findings described in this manuscript are:

1) that hypoxia acts (predominantly) via HIF-1 to inhibit RTK/RAS/MAPK-induced differentiation;

2) That HIF-1 exerts its effects on this differentiation by activating expression of the nuclear hormone receptor NHR-57; using genetics they provide evidence suggesting that NHR-57 signalling acts in parallel to RTK/RAS/MAPK signalling;

3) That NOTCH signalling upregulates the EGL-9 prolyl hydroxylase in uncommitted VPCs, most likely by direct transcriptional activation. Higher EGL-9 activity promotes HIF-1 degradation and thereby facilitates RTK/RAS/MAPK-induced differentiation by limiting NHR-57 action in these cells;

4) that these events are occurring in the VPCs, as supported by experiments that use cell-targeted RNAi.

I really have very few comments to make. The paper is clearly written, the figures are easy to digest, and the claims made by the authors are well-supported by the data they provide. There is significant interest in the pathways they are investigating across model systems and also in the context of human disease.

I strongly support publication without further review.

Minor comments:

I thought the abstract could be improved, to make clearer to readers what the discoveries made in this study were. This could simply involve saying "Here we show..." to emphasize findings made in this paper.

Reviewer #2 (Comments to the Authors (Required)):

The manuscript describes experiments indicating an overlap in Notch and HIF pathways in the overall process of cell fate decision making along the RAS/MAPK pathways. The overall experimental approach seems thorough and carefully carried out, and I have no major concerns about the manuscript. A minor concern would be the selection of the two oxygen levels of 21% and 0.5% oxygen. Although the former is of course room air, it is not clear to me what the natural history of 0.5% oxygen is for a nematode. Some justification for choosing this level, ideally coupled to experiments at some other, more intermediate oxygen levels, would be valuable.

Reviewer #3 (Comments to the Authors (Required)):

This study by Maxeiner and colleagues introduces a new signaling component to the classic system of fate patterning of the *C. elegans* Vulval Precursor Cells. The authors cleverly extrapolate from previous findings from strain hybrids, as well as drawing from diverse data sets in the literature, to examine the effects of hypoxia on VPC fate signals. They conclude that *egl-9/prolyl hydroxylase* is transcriptionally regulated by LIN-12/Notch, represses HIF-1, and HIF-1 activates NHR-57, which represses "vulval fate." The authors propose mechanisms for how hypoxia alters development. The study is a nice exploitation of the strengths of the system, and the findings are mechanistically appealing and moderately impactful.

However, methodological details and experimental and statistical abnormalities raise concerns

about the validity of the study. The authors show a confusing tendency not to distinguish between P6.p and P5/7.p in the patterning process, nor the cell fates they typically assume. For example, the terms "1°" and "2°" are rarely used. The model in figure 5 shows, under normoxic conditions, LAG-2 and LIN-12 signaling both from P6.p to P5.p and from P5.p to P6.p. We know this is not the case: LAG-2 is produced only by P6.p. Similarly, LIN-12/Notch is shown activating egl-9 expression in P6.p.

Further, there is apparently confusion about when VPC fates are induced, with a strong focus on the L2 stage. While recent results implicate some sort of signaling activity in the L2, fate reporters, notably from the Greenwald lab but also the Hajnal lab, do not come on until well into the L3 stage. Furthermore, classic analyses of T-shifts and lineage analysis implicate the L3 as the period when VPCs fates are induced. In this context, we are told that egl-9::gfp expression is not uniform among the VPCs in the L2. This would provide the first such example, since these cells are generally considered to be naïve and at least roughly equipotent. So serious concerns pertain about staging. These results also impact the reliability of the ERK-nKTR reporter, though these MPK-1 substrate results were so weak and inconsequential as to not matter much. But this doubt cast on key results for the model seriously weaken the conclusions drawn from the study.

Major points:

- 1) The authors should explain staging inconsistencies with the rest of the literature, and how these impact assays whose results are unexpected. Are they really scoring L2s? Do they know this through time? Animal size? Did the authors look at any internal developmental markers of stage, like the gonad? This issue clouds much of the interpretation
- 2) For the most part, this is a well written manuscript, with clear communication and logic. In spots conclusions are too strongly stated, and in other spots language is imprecise. Importantly imprecise language is used periodically, and, as noted, the ambiguity of fate-promoting signals is concerning.
- 3) The words "competency" and "competent" are used in the abstract and other places in the manuscript. But this term has a precise meaning in VPC patterning specifically, and in developmental biology in general. The authors did nothing to address competency, so this usage is far off base. We would be much better served by referring to these molecules as modifiers of signaling and or fate specification. We know no more than that (nor do we need to for the conclusions made in this manuscript). Imprecision in language arises in many other points. I have named them when possible, but this is by no means comprehensive. Wording is too assertive when speculative, and too tentative when the conclusions are strong. The authors should carefully parse these statements. There are also vague terms used, like "Cross-talk" both in the title and abstract.
- 4) No mention is made of the infamous drift of the let-60(n1046) (G13E) background. Are results obtained with this tool valid? At this point, the authors need to address this issue and how they deal with it. The issue is confounded by introducing the same mutation into N2 and CB4956 backgrounds. Does the Muv phenotype of these still drift? If so, how do the authors control for this phenomenon? Can they really compare between N2 and CB4856 backgrounds? What about mutant analysis in Fig. 2C and 2D?
- 5) Fig 2E-J: Why didn't the authors look at the double of egl-9(lf) with the stronger hif-1(zh111lf) with ERK-nKTR? Also with this reporter, do they want to comment on the potential shifts in the P7 and P8 mpk-1 activity.
- 6) Significance throughout was determined by bootstrapping. This is not the correct statistical analysis for these data. Further, there is no precedent for use of bootstrapping for this kind of data. Bootstrapping is typically used. Error bars of 95% confidence are also atypical.
- 7) Data presentation of Vulval induction is atypical. Do the authors justify this? A major departure from precedent should require some explanation.
- 8) I am skeptical about the use of the ERK-nKTR reporter for MPK-1 substrate activation. The authors note that the data are marginally significant. That is putting it mildly. One wonders if this

figure isn't why bootstrapping is used. I note that in the original de la Cova paper, one-way ANOVA is used as the statistical test for significance.

9) Fig 3: Corresponding DIC for the images here feels necessary, as I cannot tell staging with just the fluorescence.

10) Again the way the present the data in Fig 3B makes hard to tell the n1046 baselines. Really don't like this: if we cannot determine the baseline of a strain known to drift, how can we trust double mutant analysis? This is particularly true when builds a double mutant, thus "re-setting" the drift back to baseline, but the original single mutant strain could have drifted.

11)

In conclusion, the confusion of L2 and L3 in scoring undermines my confidence in the results, outside of the well-done genetic interactions. The authors should rectify this to convince this reviewer that these results are sound and that the model and claims in the title and abstract are supported. Other irregularities throughout, including compared to similar publications in the same field from the same lab in the past, undermine confidence in the data and the conclusions reached. I want to like the conclusions reached. But because of constant irregularities and inaccuracies I doubt the data.

Also:

Formatting: This is important: The authors say "C. elegans strains" and then list alleles used, not strains. But beyond being inaccurate, the convention now is to provide precise genotypes of all strains (not just lists of alleles) in a Supplemental Table. I must insist on this. We no longer describe how strains are made, but the SI Tables enable us to precisely explicate the strains used for the study. In a genetic model organism, this is of critical importance. Whether the authors wish to add a table with alleles/tools and origin (e.g. "this study" or "Mitani") is up to them.

"custom-made ImageJ and cell profiler scripts": These should be provided via HTML link or in a public repository

Minor improvements can be made in these places:

- Abstract: "modulates the competence": be specific. Also, "cross talk" can also be vague. These non-specific terms make it difficult to understand the abstract on a single reading
- "RAS/MAPK signaling in the three proximal VPCs ... induces the 1° fate..." Restate, as this is inaccurate
- "maintains the VPCs competent" should be "maintains VPC competency"
- Activating (gain-of-function) mutations in the RAS/MAPK pathway lead to the ectopic induction of distal VPCs and a Multivulva (Muv) phenotype, while loss-of-function mutations in RAS/MAPK pathway components cause a Vulvaless (Vul) phenotype. Thus, the average number of induced VPCs per animal, termed the vulval induction index (VI), can be used to quantify RAS/MAPK signaling strength." Is this the appropriate way to discuss this, without saying that the MPK-1 cascade induces 1° fate?
- "... (statistically marginally significant) increase in MPK-1 activity in P6.p. We conclude..." How about "We hypothesize" or even "speculate" would be better, given the weakness of this result. Also, given other concerns about staging (see above), how well were these staged for this assay?

- Fig 3E-G need better labels on the graphs to tell you are looking at egl-9:GFP signal intensity.
 - "equally expressed" would be better as "uniformly expressed"
 - "into let-60(n2021rf), lin-12 notch(n137n720lf) and lin-12 notch(n137gf) mutants" I do not see the need for "notch" here, plus we see "notch" here and "NOTCH" elsewhere. But Notch should also be: Notch. In Drosophila, alleles originally defined by a dominant mutation have the first letter capitalized. But not all letters. Best to go with "Notch."
 - Fig 2: "egl9" should be "egl-9"
 - Egl-9 as LIN-12 target gene: this paragraph has issues about strength of conclusions, varying from too strong to too weak. I recommend listing results, and then finish the paragraph with something like "Taken together" or "Collectively" followed by a strong but not absolute statement.
 - "Thus, NHR-57 is a critical HIF-1 target gene that inhibits." This should be toned down. "Target" suggests direct, which one can never determine from microarray/transcriptome analyses. "Critical" is also too strong. Why not just use "NHR-57 functions"?
 - "Our results so far have indicated" really need to moderate the conclusions. I recommend "are consistent with"
 - "This capability to adapt ultimately results in increased developmental robustness." Again, too strong. How about "may contribute" or "likely contributes to developmental robustness"?
 - "In the following, we have focused" Inappropriate in this context
 - "known transcriptional HIF-1 target genes" Use "putative" or "genes regulated by HIF-1"
 - "At the beginning of vulval induction, RAS/MAPK signaling is activated in the proximal VPCs (P5.p, P6.p and P7.p) by the LIN-3 EGF signal secreted from the AC. RAS/MAPK signaling then induces the expression of multiple DELTA family NOTCH ligands (Chen & Greenwald, 2004)." As in the Intro, key elements of sequential induction are missing from this description. Is this intentional?
 - "(la Cova et al, 2017). Should be de la Cova
 - "surrounded by a metal ring equipped with an immersion to fit in an O-ring" Indentation instead of immersion?
 - "C. elegans Genetic Center" Please use the acknowledgement wording provided by the CGC, including their funding source. This helps maintain funding for this critically important resource
-
- Nomenclature throughout: in genotypes, modifiers should be non-italicized while the rest of the gene and alleles name are italicized. Example: let-23(sy1rf)
 - Nomenclature: shouldn't zh121 and zh122 have "gf" after them?
 - Nomenclature: "bar-1 β -catenin(ga80)" is non-standard
 - Nomenclature: "lin-12(n137n720lf)" If used in some places and rf in others. Likewise with egl-9. Please be consistent.
 - Nomenclature: "salS14" should be "sals14"

To Reviewer #1:

Improving the abstract

"I thought the abstract could be improved, to make clearer to readers what the discoveries made in this study were. This could simply involve saying "Here we show..." to emphasize findings made in this paper."

We have changed abstract to indicate where the new finding begin and tried to improve. clarity overall. See changes p.2.

To Reviewer #2:

Use of 0.5% oxygen for hypoxia experiments

"... it is not clear to me what the natural history of 0.5% oxygen is for a nematode. Some justification for choosing this level, ideally coupled to experiments at some other, more intermediate oxygen levels, would be valuable."

0.5% is the lowest oxygen level at which the animals develop. This point is now explained on p. 5, 1st para.

To Reviewer #3:

1. Distinction between 1° and 2° fates

"The authors show a confusing tendency not to distinguish between P6.p and P5/7.p in the patterning process, nor the cell fates they typically assume. For example, the terms "1°" and "2°" are rarely used."

Yes, we did not distinguish 1° and 2° fates because the focus was on induction of vulval differentiation rather than on fate patterning. We found no evidence that oxygen levels are directly involved in making the distinction between the 1° vs 2° fates. Hypoxia affects RAS/MAPK but not LIN-12 pathway mutants (fig. 1D). Since RAS/MAPK signaling is required to activate LIN-12 NOTCH signaling, hypoxia probably also reduces NOTCH activation. This point is now better explained in the introduction and also in the modified discussion of our model (p. 12 2nd para).

2. When are the LIN-3 EGF and LIN-12 pathways first activated?"

"The model in figure 5 shows, under normoxic conditions, LAG-2 and LIN-12 signaling both from P6.p to P5.p and from P5.p to P6.p. We know this is not the case: LAG-2 is produced only by P6.p. Similarly, LIN-12/Notch is shown activating egl-9 expression in P6.p."

Similar to the RAS/MAPK pathway (see below point 3), there exists strong evidence indicating that the Notch signaling pathway is already activated during the L2 fate. For example, Chen and Greenwald (2004) found that the Notch ligand LAG-2 is initially expressed in all VPCs of L2 larvae (see fig.5b in that article), while Leviatan et al. 1998 showed that also LIN-12 NOTCH is expressed in all VPCs until the late-L2/early L3 stage (see Fig 4 in Leviatan et al.), when LIN-12 is down-regulated in

P6.p. Similarly, Nusser-Stein et al (2012) observed LIN-12 signaling activity in G1 phase VPCs of L2 stage larvae, based on the expression pattern of Notch reporters and of the Notch target gene *lip-1*, which is already expressed in L2 larvae (Berset et al. 2001). Hence, our model that EGL-9 expression is induced by Notch signaling already during the L2 stage to regulate the competence of the VPCs (i.e. their capability to respond to inductive signaling) is based on published data about Notch signaling in the VPCs of L2 larvae.

3. When are the VPC fates induced?

“Further, there is apparently confusion about when VPC fates are induced, with a strong focus on the L2 stage. While recent results implicate some sort of signaling activity in the L2, fate reporters, notably from the Greenwald lab but also the Hajnal lab, do not come on until well into the L3 stage.”

And

*“Furthermore, classic analyses of T-shifts and lineage analysis implicate the L3 as the period when VPCs fates are induced. In this context, we are told that *egl-9::gfp* expression is not uniform among the VPCs in the L2. This would provide the first such example, since these cells are generally considered to be naïve and at least roughly equipotent.”*

The current view is that vulval induction does not take place at a certain time point of development but rather occurs over a specific time period, beginning in the early L2 stage as soon as the anchor cell is born and starts expressing the LIN-3 EGF growth factor. The VPC fates are irreversibly determined after the first round of divisions in mid-L3 larvae, once the 3° VPCs have fused with *hyp7*. Until that time point, 2° and even 3° VPCs can still be converted into 1° cells by a pulse of MPK-1 activity (Wang & Sternberg 1999, Ambros 1999).

Analogous to Notch signaling (see above point 2.), there exists good evidence that the EGFR/RAS/MAPK signaling pathway is activated already during the mid-L2 stage and remains active even after fate specification during vulval morphogenesis in L3 and L4 larvae. This conclusion is not only based on the expression pattern of RAS/MAPK and Notch target genes, such as *lin-39* (Maloof and Kenyon, 1998) *egl-17* (Burdine(1998) and *lip-1* (Berset, 2001), which all exhibit a non-uniform expression pattern in the VPCs of L2 larvae, but also on the ERK biosensor that demonstrated high RAS/MAPK activity in P6.p of mid-L2 larvae (de laCova, 2017, Fig. 2 and p.544 in that article). Thus, to examine the effect of the hypoxia pathway on EGFR/RAS/MAPK signaling, the mid-L2 stage is the critical stage to analyze.

4. Staging of the animals for the *hif-1 egl-9* and *nhr-57* reporter analysis

“Major points:

The authors should explain staging inconsistencies with the rest of the literature, and how these impact assays whose results are unexpected. Are they really scoring L2s? Do they know this through time? Animal size? Did the authors look at any internal developmental markers of stage, like the gonad? This issue clouds much of the interpretation.”

And

“9) Fig 3: Corresponding DIC for the images here feels necessary, as I cannot tell staging with just the fluorescence.”

And

“11) In conclusion, the confusion of L2 and L3 in scoring undermines my confidence in the results, outside of the well-done genetic interactions. The authors should rectify this to convince this reviewer that these results are sound and that the model and claims in the title and abstract are supported. Other irregularities throughout, including compared to similar publications in the same field from the same lab in the past, undermine confidence in the data and the conclusions reached. I want to like the conclusions reached. But because of constant irregularities and inaccuracies I doubt the data.”

These comments refer to the staging of the animals shown in Fig 3A, top panel, in 3C, top two panels, and in Fig 4D., top panels. All others animals are at the Pn.px stage or later and thus mid-L3 larvae or older.

Our method of synchronization and staging is now explained in the Materials section (p. 14, last para). In brief, we produced synchronized populations of L2 or L3 larvae by re-feeding starvation-arrested L1 larvae for specific time periods. To further distinguish the stages, we examined, besides scoring the VPC divisions, the gonad morphology (i.e. dorsal turning of the DTC, which occurs during or shortly after the L2/L3 molt) and measured the gonad length to distinguish mid-L2 from late-L2/early L3 larvae.

For example, in the animal shown in Fig. 3C, the proximal VPCs were undivided, while P8.p and P4.p had just divided (see screenshots below). The DTCs had not yet turned and the gonad length was 111 μ m. Hence, we scored this animal as late-L2/early-L3 larva.

The top panel in Fig. 3C now shows a mid-L2 larva (gonad length 70 μ m), a slightly earlier stage when the bias in proximal vs. distal EGL-9::GFP first becomes visible. See also the modified text of this section on p.8, 2nd para.

5. Staging of the animals for MPK-1 biosensor analysis

“So serious concerns pertain about staging. These results also impact the reliability of the ERK-nKTR reporter, though these MPK-1 substrate results were so weak and inconsequential as to not matter much. But this doubt cast on key results for the model seriously weaken the conclusions drawn from the study.”

We used the same criteria for staging as described above under point 4. As explained above under point 3., we wanted to focus on the mid-L2 stage. To further strengthen our results, we have re-analyzed the data by plotting ERK sensor activity in P6.p versus gonad length (see graph below). While there seems to be no strong correlation between gonad length and activity, we realized that some of the animals originally included were rather late- than mid-L2 larvae and we removed them from our analysis (i.e. we filtered the data for gonad length <75 μ m). This modification to the analysis does not change our conclusion that the hypoxia pathway does not directly alter the activity of the RAS/MAPK pathway, but rather modifies the response of the VPCs to MAPK activation. The previously weakly significant increase in *egl-9 hif-1* double mutants is now insignificant (see also below point 10 on the statistical analysis method used). We believe, these are important data that should be shown in Fig. 2 of the manuscript, even though it is a “negative” result.

6. Use of the terms competency and competent

“3) The words “competency” and “competent” are used in the abstract and other places in the manuscript. But this term has a precise meaning in VPC patterning specifically, and in developmental biology in general. The authors did nothing to address competency, so this usage is far off base.”

I suppose the reviewer is referring to our use of the term “competence”, since we have not used the term “competency” in our manuscript. This term (and the

adjective “competent”) is commonly used when referring to the capability of the VPCs to respond to inductive and lateral signaling. To avoid any confusion, we have modified the abstract (p.2) and define our use of this term in the introduction (p.3, 1st para).

7. Improving imprecise language

“2) For the most part, this is a well written manuscript, with clear communication and logic. In spots conclusions are too strongly stated, and in other spots language is imprecise. Importantly imprecise language is used periodically, and, as noted, the ambiguity of fate-promoting signals is concerning.”

And

“We would be much better served by referring to these molecules as modifiers of signaling and or fate specification. We know no more than that (nor do we need to for the conclusions made in this manuscript). Imprecision in language arises in many other points. I have named them when possible, but this is by no means comprehensive. Wording is too assertive when speculative, and too tentative when the conclusions are strong. The authors should carefully parse these statements. There are also vague terms used, like “Cross-talk” both in the title and abstract.”

We have revised the manuscript throughout to avoid any misunderstanding (see marked-up version). The term “cross-talk” is widely used when referring to the interaction between two signaling pathways, and we would like to keep it in the manuscript. However, we have shortened the title to “The hypoxia-response pathway modulates RAS/MAPK-mediated cell fate decisions in *C. elegans*”

8. How to address the drift of the *let-60(n1046)* allele

*“4) No mention is made of the infamous drift of the *let-60(n1046)* (G13E) background. Are results obtained with this tool valid? At this point, the authors need to address this issue and how they deal with it.”*

And

“The issue is confounded by introducing the same mutation into N2 and CB4956 backgrounds. Does the Muv phenotype of these still drift? If so, how do the authors control for this phenomenon? Can they really compare between N2 and CB4856 backgrounds? What about mutant analysis in Fig. 2C and 2D?”

We are well aware of the drift of the *n1046* background. As indicated already in the first version of the manuscript (p. 6, 1st para), we scored control *n1046* siblings obtained from in each cross with a hypoxia pathway mutant (Fig. 2C). In our experience, this is the appropriate control for effects of the genetic background. The specific control strains used are now listed in suppl. Table S3. For the data in Fig 2D, genetic drift is not an issue since we compared siblings of the same genetic background grown on the same plates with and without *egl-9* rescue transgenes. Also, the difference between CRISPR/CAS9-induced alleles is unlikely to be

caused by genetic drift as they were recently generated in our lab and kept as frozen stocks.

7. ERK sensor activity in *egl-9(lf) hif-1(zh111)* background and did we observe shifts in MPK-1 activity?

*“5) Fig 2E-J: Why didn't the authors look at the double of *egl-9(lf)* with the stronger *hif-1(zh111lf)* with ERK-nKTR? Also with this reporter, do they want to comment on the potential shifts in the P7 and P8 *mpk-1* activity.”*

We did not examine the double *egl-9(lf)* mutant with *zh111* because the *zh111* single mutant alone did not show a change in MPK-1 activity. We did not observe any obvious shifts, i.e. P6.p showed highest MPK-1 activity in all but a few cases (Fig. 2E-I).

8. Statistical analysis: Bootstrapping vs. t-tests

“6) Significance throughout was determined by bootstrapping. This is not the correct statistical analysis for these data.”

And

“Further, there is no precedent for use of bootstrapping for this kind of data. Bootstrapping is typically used. Error bars of 95% confidence are also atypical.”

After consulting with two statisticians for a previous publication, we learned that that a two-tailed t-test, even though commonly used, is not the correct test, since vulval induction counts are based on discrete values and hence cannot be normally distributed. We were advised to perform non-parametric tests or use bootstrap analysis. We have been routinely using bootstrap analysis of vulval induction counts (see for example Walser et al. 2017). To further address this issue, we have compared the results of our analysis by bootstrapping to two-tailed t-tests of the same dataset (see the data analysis in suppl. Table S4 with the raw data). Note that a p-value of 0 in a bootstrap analysis of 1000 samples signifies $p \leq 0.001$. We found no relevant differences between the two methods (i.e. there is no case, where a significant results was obtained by bootstrapping that was insignificant in a t-test, or vice versa). Thus, we prefer to use bootstrapping for the reasons mentioned above.

Most studies use a p -value < 0.05 as threshold for statistical significance. This threshold can also be represented by 95% confidence intervals. The standard deviations can be found in suppl. Table S4.

9. Presentation of the vulval induction data

“7) Data presentation of Vulval induction is atypical. Do the authors justify this? A major departure from precedent should require some explanation.”

We compared both options of data presentation and found the option of showing the ΔVI values together with the absolute VI for the condition tested (e.g. hypoxia) to be easier to understand. All raw data and the absolute VI values for both conditions can be found in suppl. Table S4, such that a reader can exactly reproduce how the analysis was done.

10. Statistical analysis of the ERK sensor data

“8) I am skeptical about the use of the ERK-nKTR reporter for MPK-1 substrate activation. The authors note that the data are marginally significant. That is putting it mildly. One wonders if this figure isn't why bootstrapping is used. I note that in the original de la Cova paper, one-way ANOVA is used as the statistical test for significance.”

We forgot to mention in the original version that the analysis of the ERK sensor data was done by ANOVA with an appropriate multiple comparison correction. This is now mentioned on P.7, 2nd para and on p.15, last para.

11. Presentation of tissue-specific RNAi data in Fig. 3B

“10) Again the way the present the data in Fig 3B makes hard to tell the n1046 baselines. Really don't like this: if we cannot determine the baseline of a strain known to drift, how can we trust double mutant analysis? This is particularly true when builds a double mutant, thus "re-setting" the drift back to baseline, but the original single mutant strain could have drifted.”

Genetic drift is not an issue in this experiment because the same genetic background was compared after feeding two different RNAi bacterial strains (empty vector vs. hif-1 RNAi). However, as mentioned in the original version on p. 8 1st para, the Pn.p and gut-specific RNAi strains do exhibit a different baseline of the n1046 phenotype, which is probably caused by the different *rde-1* transgenes used (gut- vs. Pn.p-specific expression).

12. Strains and alleles used

“Formatting: This is important: The authors say "C. elegans strains" and then list alleles used, not strains. But beyond being inaccurate, the convention now is to provide precise genotypes of all strains (not just lists of alleles) in a Supplemental Table. I must insist on this. We no longer describe how strains are made, but the SI Tables enable us to precisely explicate the strains used for the study. In a genetic model organism, this is of critical importance Whether the authors wish to add a table with alleles/tools and origin (e.g. "this study" or "Mitani") is up to them.”

Table S3 now includes a list of all strains used, including the control siblings analyzed for the *let-60(n1046)* crosses. We changed “strains” to “alleles” in the methods section (p.14, top).

13. Quantification of MPK-1 biosensor

"custom-made ImageJ and cell profiler scripts": These should be provided via HTML link or in a public repository”

The codes of the Image J and cell profiler scripts are now included in the supplemental information. The codes can easily be added to an existing image processing pipeline, though the file naming and other parameters have to be adapted to the specific imaging system used.

14. Minor points

“Minor improvements can be made in these places:

- *Abstract: "modulates the competence": be specific. Also, "cross talk" can also be vague. These non-specific terms make it difficult to understand the abstract on a single reading “*

The abstract has been modified as explained above under points 6 And 7.

See above point 6.

- *"RAS/MAPK signaling in the three proximal VPCs ... induces the 1° fate..."
Restate, as this is inaccurate*

This section of the introduction has been modified accordingly, see p.3, 1st para.

- *"maintains the VPCs competent" should be "maintains VPC competency"*

See above point 6.

“• Activating (gain-of-function) mutations in the RAS/MAPK pathway lead to the ectopic induction of distal VPCs and a Multivulva (Muv) phenotype, while loss-of-function mutations in RAS/MAPK pathway components cause a Vulvaless (Vul) phenotype. Thus, the average number of induced VPCs per animal, termed the vulval induction index (VI), can be used to quantify RAS/MAPK signaling strength. Is this the appropriate way to discuss this, without saying that the MPK-1 cascade induces 1° fate?”

The modified introduction (p.3, 1st para) now contains a better explanation of the sequential induction model.

- *"...(statistically marginally significant) increase in MPK-1 activity in P6.p. We conclude..." How about "We hypothesize" or even "speculate" would be better, given the weakness of this result. Also, given other concerns about staging (see above), how well were these staged for this assay?*

As explained above under point 5, after including only mid-L2 larvae in the analysis, we find no statistically significant differences in the hypoxia mutants compared to the wild-type. See also the modified text on p.7, 2nd para.

- *Fig 3E-G need better labels on the graphs to tell you are looking at egl-9:GFP signal intensity.*

Corrected

- *"equally expressed" would be better as "uniformly expressed"*

Corrected, see p.8, 2nd para.

- *"into let-60(n2021rf), lin-12 notch(n137n720lf) and lin-12 notch(n137gf) mutants" I do not see the need for "notch" here, plus we see "notch" here and "NOTCH" elsewhere. But Notch should also be: Notch. In Drosophila, alleles originally defined by a dominant mutation have the first letter capitalized. But not all letters. Best to go with "Notch."*

Corrected

- *Fig 2: "egl9" should be "egl-9"*

Corrected

- *Egl-9 as LIN-12 target gene: this paragraph has issues about strength of conclusions, varying from too strong to too weak. I recommend listing results, and then finish the paragraph with something like "Taken together" or "Collectively" followed by a strong but not absolute statement.*

We rearranged this paragraph to better split results and interpretation (p.8, 3d para).

- *"Thus, NHR-57 is a critical HIF-1 target gene that inhibits." This should be toned down. "Target" suggests direct, which one can never determine from microarray/transcriptome analyses. "Critical" is also too strong. Why not just use "NHR-57 functions"?*

nhr-57 is commonly referred to in the literature as a HIF-1 target gene (e.g. Shen et al. 2005). Since the *nhr-57* locus contains 4 HREs, it is likely a direct HIF-1 target. Moreover, we deleted this sentence (last para on p.9) since it was redundant with the conclusion of this section (p.10, 1st para).

- *"Our results so far have indicated" really need to moderate the conclusions. I recommend "are consistent with"*

Corrected ("suggested")

- *"This capability to adapt ultimately results in increased developmental robustness." Again, too strong. How about "may contribute" or "likely contributes to developmental robustness"?*

Corrected ("could")

- *"In the following, we have focused" Inappropriate in this context*

Corrected

- *"known transcriptional HIF-1 target genes" Use "putative" or "genes regulated by HIF-1"*

Corrected ("...a selection of candidate genes regulated by HIF-1")

- *"At the beginning of vulval induction, RAS/MAPK signaling is activated in the proximal VPCs (P5.p, P6.p and P7.p) by the LIN-3 EGF signal secreted from the AC. RAS/MAPK signaling then induces the expression of multiple DELTA family NOTCH ligands (Chen & Greenwald, 2004)." As in the Intro, key elements of sequential induction are missing from this description. Is this intentional?*
We have modified this paragraph to make our model cleared (p. 12, 2nd para).

- *"(la Cova et al, 2017). Should be de la Cova*

Corrected

- *"surrounded by a metal ring equipped with an immersion to fit in an O-ring"*
Indentation instead of immersion?

Corrected

- *"C. elegans Genetic Center" Please use the acknowledgement wording provided by the CGC, including their funding source. This helps maintain funding for this critically important resource*

Corrected

- *Nomenclature throughout: in genotypes, modifiers should be non-italicized while the rest of the gene and alleles name are italicized. Example: let-23(sy1rf)*
- *Nomenclature: shouldn't zh121 and zh122 have "gf" after them?*

Corrected

- *Nomenclature: "bar-1 β -catenin(ga80)" is non-standard*

Corrected

- *Nomenclature: "lin-12(n137n720lf)" If used in some places and rf in others. Likewise with egl-9. Please be consistent.*

We only used rf for the let-60(n2021) allele because it's a hypomorph, all others are lf.

- *Nomenclature: "saIS14" should be "sals14" "*

Corrected

May 13, 2019

RE: Life Science Alliance Manuscript #LSA-2018-00255-TR

Dr. Alex Hajnal
University of Zurich
Institute of Molecular Life Sciences
Winterthurerstr. 190
Zurich CH-8057
Switzerland

Dear Dr. Hajnal,

Thank you for submitting your revised manuscript entitled "The hypoxia-response pathway modulates RAS/MAPK-mediated cell fate decisions in *C. elegans*". As you will see, the reviewers appreciate the introduced changes though reviewer #3 calls out some mis-representations of the literature (in your point-by-point response). We would be thus happy to publish your paper in Life Science Alliance pending final revisions:

- please check your introductory parts on NOTCH/NOTCH ligands carefully in light of reviewer #3's remarks
- we display suppl figures in-line in the HTML version of the paper => please upload the supplementary figures as individual files and without legends, the legends can be added to the main manuscript file; please upload the ImageJ and CellProfiler scripts as supplementary material / text.

A. FINAL FILES:

- An editable version of the final text (.DOC or .DOCX) is needed for copyediting (no PDFs).
- High-resolution figure, supplementary figure and video files uploaded as individual files: See our detailed guidelines for preparing your production-ready images, <http://www.life-science-alliance.org/authors>
- Summary blurb (enter in submission system): A short text summarizing in a single sentence the

study (max. 200 characters including spaces). This text is used in conjunction with the titles of papers, hence should be informative and complementary to the title. It should describe the context and significance of the findings for a general readership; it should be written in the present tense and refer to the work in the third person. Author names should not be mentioned.

B. MANUSCRIPT ORGANIZATION AND FORMATTING:

Sincerely,

Andrea Leibfried, PhD
Executive Editor
Life Science Alliance
Meyrhofstr. 1
69117 Heidelberg, Germany
t +49 6221 8891 502
e a.leibfried@life-science-alliance.org
www.life-science-alliance.org

Reviewer #1 (Comments to the Authors (Required)):

I am satisfied with how the authors have addressed both my comments and those of the other referees. I recommend publication of the manuscript.

Reviewer #3 (Comments to the Authors (Required)):

The authors provide a detailed rebuttal of critiques presented in review. I don't want to be that reviewer who holds up interesting observations from being published. So I will not do so. But I do feel it is incumbent on me to note mis-referencing in the rebuttal.

Since the points are not central, I will not insist on their being fixed. But I also will not let inaccuracies pass unchallenged.

The authors stated in their rebuttal: "For example, Chen and Greenwald (2004) found that the Notch ligand LAG-2 is initially expressed in all VPCs of L2 larvae (see fig.5b in that article),"

But this is not true! The relevant text from Chen and Greenwald (2004) says, "Expression of the transcriptional reporter for lag-2 is evident in all six VPCs in the early L3 stage (Figure 5B)...." Chen and Greenwald go on to note, "The restricted expression or upregulation of apx-1, dsl-1, and lag-2 in P6.p strongly supports the hypothesis that the lateral signal originates in the presumptive 1 VPC and directly activates LIN-12 in the neighboring VPCs."

Likewise, Notch DSL ligands APX-1 and DSL-1 are similarly not expressed in early L3s (see the paragraph above: this developmental point is not estimated by time from starvation-synchronization, but by length of extension of the gonadal distal tip cells).

Similarly, Levitan et al 1998 is also misquoted. The authors in their rebuttal state, "while Leviatan et al. 1998 showed that also LIN-12 NOTCH is expressed in all VPCs until the late-L2/early L3 stage (see Fig 4 in Leviatan et al.), when LIN-12 is down-regulated in P6.p." Yet looking at Levitan, I found: "The level of lin-12::lacZ expression from the early L2 stage until the VPCs divide in the L3 stage appears to be uniform in all six VPCs (Wilkinson and Greenwald, 1995). ... in the mid-L3 stage, at the time of VPC specification, LIN-12::GFP is reduced in P6.p relative to the other VPCs...."

We all want to marshal evidence to support our arguments. But it is critical that such citations be correct! I'll also note that expression of receptor does not mean that signaling is occurring. Expression of the ligand, at least in this case, likely does so. This may be quibbling, and does not alter the overall observations of the manuscript. But precision is important.

I think I understand the points made by the authors. I just don't think they were clearly made in the manuscript. Yet these points do not undermine the interesting data, which we should allow to be published.

May 17, 2019

RE: Life Science Alliance Manuscript #LSA-2018-00255-TRR

Dr. Alex Hajnal
University of Zurich
Institute of Molecular Life Sciences
Winterthurerstr. 190
Zurich CH-8057
Switzerland

Dear Dr. Hajnal,

Thank you for submitting your Research Article entitled "The hypoxia-response pathway modulates RAS/MAPK-mediated cell fate decisions in *C. elegans*". It is a pleasure to let you know that your manuscript is now accepted for publication in Life Science Alliance. Congratulations on this interesting work.

DISTRIBUTION OF MATERIALS:

Again, congratulations on a very nice paper. I hope you found the review process to be constructive and are pleased with how the manuscript was handled editorially. We look forward to future exciting submissions from your lab.

Sincerely,
